# CONSTRUCTING SEMANTICS-AWARE ADVERSARIAL EXAMPLES WITH PROBABILISTIC PERSPECTIVE

## ABSTRACT

In this study, we introduce a novel, probabilistic perspective for generating adversarial examples. Within this view, geometric constraints on adversarial examples are interpreted as distributions, facilitating the transition from geometric constraints to data-driven semantic constraints. Proceeding from this perspective, we develop an innovative approach for generating semantics-aware adversarial examples in a principled manner. Our approach empowers individuals to incorporate their personal comprehension of semantics into the model. Through human evaluation, we validate that our semantics-aware adversarial examples maintain their inherent meaning. Experimental findings on the MNIST, SVHN and CIFAR10 datasets demonstrate that our semantics-aware adversarial examples can effectively circumvent robust adversarial training methods tailored for traditional adversarial attacks.

## 1 INTRODUCTION

The purpose of generating adversarial examples is to deceive a classifier by making minimal changes to the original data's meaning. In image classification, most existing adversarial techniques ensure the preservation of adversarial example semantics by limiting their geometric distance from the original image (Szegedy et al., 2013; Goodfellow et al., 2014; Carlini & Wagner, 2017; Madry et al., 2017). These methods are able to deceive classifiers with a very small geometric based perturbation. However, when targeting robust classifiers trained using adversarial methods, an attack involving a relatively large geometric distance may be necessary. Such alterations have a notable drawback: they can either distort the original image's semantics - going against the fundamental objective of adversarial examples - or lead to discernible changes in the image. Figure 1 illustrates this issue. When applying the PGD attack, constrained by geometric distances, to robust classifiers for digit images, the resulting adversarial images frequently deviate from their original semantic intent. In the case of natural images, even if the core semantics are preserved, the introduced changes can often be readily observed.

In this paper, we introduce a probabilistic perspective for understanding adversarial examples. Through this innovative lens, both the victim classifier and geometric constraints are regarded as distinct distributions: the victim distribution and the distance distribution. Adversarial examples emerge as samples drawn from the product of these two distributions, specifically from the regions where they overlap. Notably, the overlap at the tail of the distance distribution accounts for the apparent modifications in the resultant adversarial samples.

Based on this probabilistic perspective, we propose a new method for generating semantics-aware adversarial examples. Instead of relying on purely geometrically-induced distance distributions, we transition to a trainable, data-driven distance distribution. This evolution allows for the incorporation of our subjective understanding of semantics. With an appropriate semantic comprehension, our method can produce adversarial examples that are not only semantically aware but also highly effective. As depicted in the right section of Figure 1, our semantics-aware adversarial attack successfully deceives robust classifiers while largely retaining the original image's semantic essence.

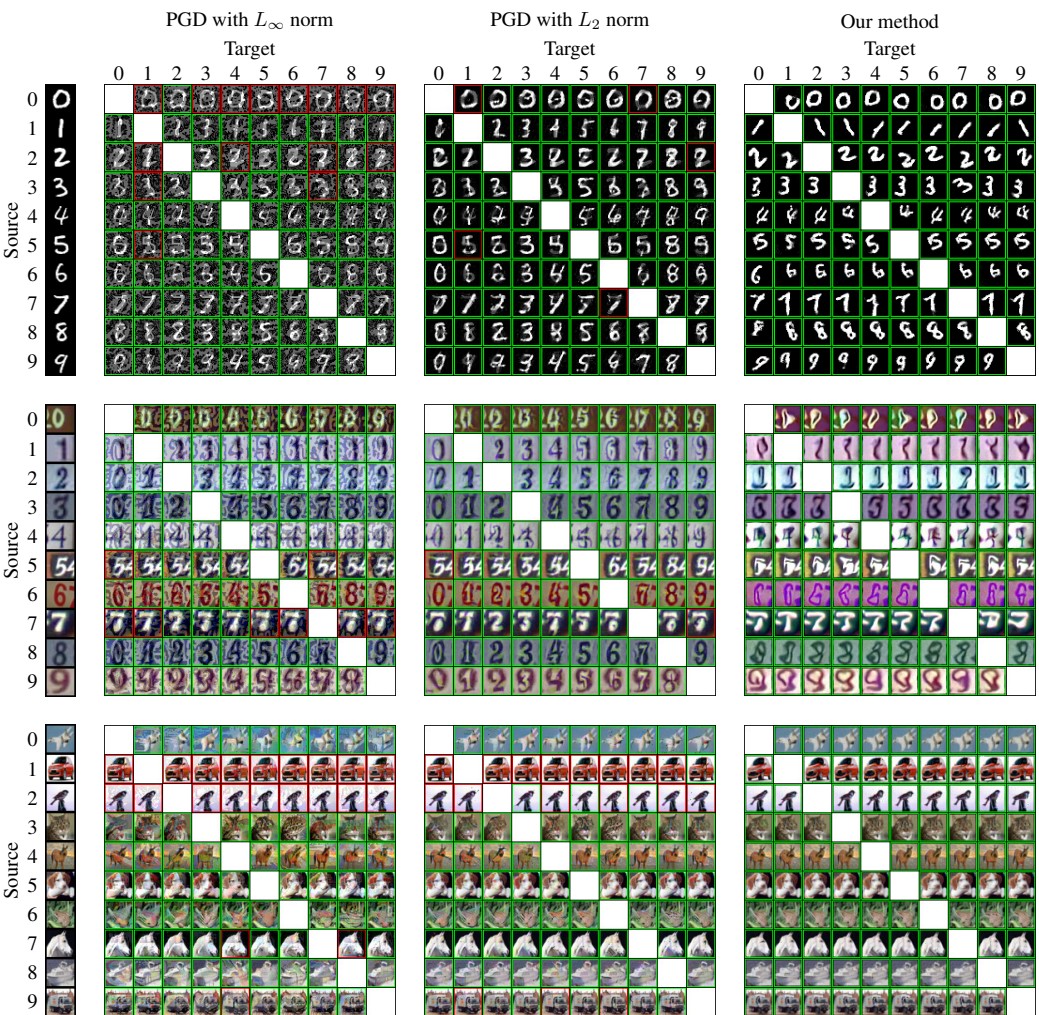

Figure 1: Attacks on adversarially trained classifiers for MNIST, SVHN, and CIFAR10: Comparing PGD with $L_2$ norm, PGD with $L_\infty$ norm, and our proposed method. Parameters are adjusted to ensure that most of the cases could successfully deceive the victim classifier. A green border marks a successful attack, while red denotes failure. See Appendix K for more examples.

## 2 PRELIMINARIES

### 2.1 ADVERSARIAL EXAMPLES

The notion of adversarial examples was first introduced by Szegedy et al. (2013). Let's assume we have a classifier $C : [0, 1]^n \to \mathcal{Y}$, where $n$ represents the dimension of the input space and $\mathcal{Y}$ denotes the label space. Given an image $\mathbf{x}_{\text{ori}} \in [0, 1]^n$ and a target label $y_{\text{tar}} \in \mathcal{Y}$, the optimization problem for finding an adversarial instance $\mathbf{x}_{\text{adv}}$ for $\mathbf{x}_{\text{ori}}$ can be formulated as follows:

$$\text{minimize } \mathcal{D}(\mathbf{x}_{\text{ori}}, \mathbf{x}_{\text{adv}}) \quad \text{such that } C(\mathbf{x}_{\text{adv}}) = y_{\text{tar}} \text{ and } \mathbf{x}_{\text{adv}} \in [0, 1]^n$$

Here, $\mathcal{D}$ is a distance metric employed to assess the difference between the original and perturbed images. This distance metric typically relies on geometric distance, which can be represented by $L_0$, $L_2$, or $L_\infty$ norms.

However, solving this problem is challenging. As a result, Szegedy et al. (2013) propose a relaxation of the problem:

$$\text{minimize } \mathcal{L}(\mathbf{x}_{\text{adv}}, y_{\text{tar}}) := c_1 \cdot \mathcal{D}(\mathbf{x}_{\text{ori}}, \mathbf{x}_{\text{adv}}) + c_2 \cdot f(\mathbf{x}_{\text{adv}}, y_{\text{tar}}) \quad \text{such that } \mathbf{x}_{\text{adv}} \in [0, 1]^n \quad (1)$$

where $c_1$, $c_2$ are constants, and $f$ is an objective function closely tied to the classifier's prediction. For example, in (Szegedy et al., 2013), $f$ is the cross-entropy loss function, while Carlini & Wagner (2017) suggest several different choices for $f$. Szegedy et al. (2013) recommend solving (1) using box-constrained L-BFGS.

## 2.2 Adversarial training

Adversarial training, a widely acknowledged method for boosting adversarial robustness in deep learning models, has been extensively studied (Szegedy et al., 2013; Goodfellow et al., 2014; Huang et al., 2015; Madry et al., 2017). This technique uses adversarial samples as (part of) the training data, originating from Szegedy et al. (2013), and has evolved into numerous variations. In this paper, we apply the min-max problem formulation by Madry et al. (2017) to determine neural network weights, denoted as $\theta$. They propose choosing $\theta$ to solve:

$$\min_{\theta} \mathbb{E}_{(\mathbf{x},y)\sim p_{\text{data}}} \left[ \max_{\|\delta\|_p \leq \epsilon} \mathcal{L}_{\text{CE}}(\theta, \mathbf{x} + \delta, y) \right] \tag{2}$$

where $p_{\text{data}}$ represents the data distribution, $\mathcal{L}_{\text{CE}}$ is the cross-entropy loss, $\|\cdot\|_p$ denotes the $L_p$ norm, and $\epsilon$ specifies the radius of the corresponding $L_p$ ball. In what follows, we will use the term "robust classifier" to refer to classifiers that have undergone adversarial training.

## 2.3 Energy-based models (EBMs)

An Energy-based Model (EBM) (Hinton, 2002; Du & Mordatch, 2019) involves a non-linear regression function, represented by $E_\theta$, with a parameter $\theta$. This function is known as the energy function. Given a data point, $\mathbf{x}$, the probability density function (PDF) is given by:

$$p_\theta(\mathbf{x}) = \frac{\exp(-E_\theta(\mathbf{x}))}{Z_\theta} \tag{3}$$

where $Z_\theta = \int \exp(-E_\theta(\mathbf{x}))\mathrm{d}\mathbf{x}$ is the normalizing constant that ensures the PDF integrates to 1. For details on sampling and training energy-based models, refer to Appendix A.

## 3 A probabilistic perspective on adversarial examples

We introduce a probabilistic perspective where adversarial examples are sampled from an adversarial distribution, denoted as $p_{\text{adv}}$. This distribution can be conceptualized as a product of expert distributions (Hinton, 2002):

$$p_{\text{adv}}(\mathbf{x}_{\text{adv}}; \mathbf{x}_{\text{ori}}, y_{\text{tar}}) \propto p_{\text{vic}}(\mathbf{x}_{\text{adv}}; y_{\text{tar}})p_{\text{dis}}(\mathbf{x}_{\text{adv}}; \mathbf{x}_{\text{ori}})p_{\text{pri}}(\mathbf{x}_{\text{adv}}) \tag{4}$$

where $p_{\text{vic}}$ is termed the "victim distribution", reflecting its association with the victim classifier. $p_{\text{dis}}$ represents the distance distribution. A substantial value of $p_{\text{dis}}$ suggests a close resemblance between $\mathbf{x}_{\text{adv}}$ and $\mathbf{x}_{\text{ori}}$ based on the designated distance metric. $p_{\text{pri}}$ acts as the prior distribution for $\mathbf{x}_{\text{adv}}$.

The subsequent theorem demonstrates the compatibility of our probabilistic approach with the conventional optimization problem for generating adversarial examples:

**Theorem.** *Given the condition that $p_{vic}(\mathbf{x}_{adv}; y_{tar}) \propto \exp(-c_2 \cdot f(\mathbf{x}_{adv}, y_{tar}))$, $p_{dis}(\mathbf{x}_{adv}; \mathbf{x}_{ori}) \propto \exp(-c_1 \cdot \mathcal{D}(\mathbf{x}_{ori}, \mathbf{x}_{adv}))$, and set $p_{pri}(\mathbf{x}_{adv})$ as a constant, the samples drawn from $p_{adv}$ will exhibit the same distribution as the adversarial examples derived from applying the box-constrained Langevin Monte Carlo method to the optimization problem delineated in equation (1).*

The proof of the theorem can be found in Appendix B. Within the context of our discussion, we initially define $p_{\text{vic}}$, $p_{\text{dis}}$, and $p_{\text{pri}}$ to have the same form as described in the theorem. Given this formulation, we can conveniently generate samples from $p_{\text{adv}}$, $p_{\text{dis}}$, and $p_{\text{vic}}$ using LMC. Detailed procedures are provided in Appendix C. As we delve further into this paper, we may explore alternative formulations for these components.

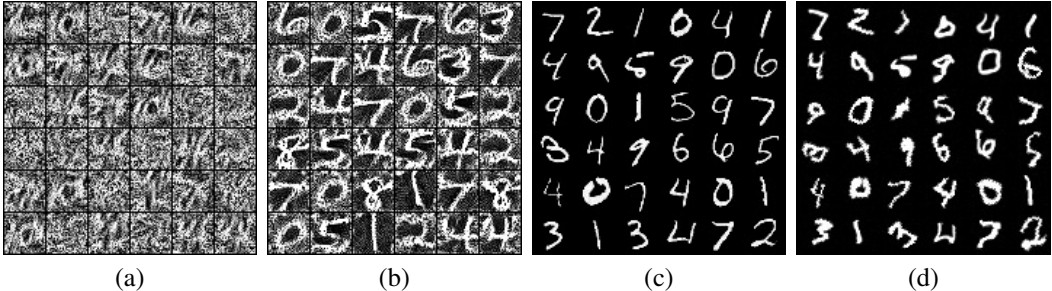

$$(a) \qquad\qquad (b) \qquad\qquad (c) \qquad\qquad (d)$$

Figure 2: **(a)** and **(b)** display samples drawn from $p_{\text{vic}}(\cdot; y_{\text{tar}})$ with the victim classifier being non-adversarially trained and adversarially trained, respectively. **(c)** showcases samples from $p_{\text{dis}}(\cdot; \mathbf{x}_{\text{ori}})$ when $\mathcal{D}$ is the square of $L_2$ norm. **(d)** illustrates $t(\mathbf{x}_{\text{ori}})$ for $t \sim \mathcal{T}$, where $\mathcal{T}$ represents a distribution of transformations, including TPS (see Section 5.2), scaling, rotation, and cropping. The $\mathbf{x}_{\text{ori}}$s in (c) and (d) consist of the first 36 images from the MNIST test set.

**The victim distribution** $p_{\text{vic}}$ is dependent on the victim classifier. As suggested by Szegedy et al. (2013), $f$ could be the cross-entropy loss of the classifier. We can sample from this distribution using Langevin dynamics. Figure 2(a) presents samples drawn from $p_{\text{vic}}$ when the victim classifier is subjected to standard training, exhibiting somewhat indistinct shapes of the digits. This implies that the classifier has learned the semantics of the digits to a certain degree, but not thoroughly. In contrast, Figure 2(b) displays samples drawn from $p_{\text{vic}}$ when the victim classifier undergoes adversarial training. In this scenario, the shapes of the digits are clearly discernible. This observation suggests that we can obtain meaningful samples from adversarially trained classifiers, indicating that such classifiers depend more on semantics, which corresponds to the fact that an adversarially trained classifier is more difficult to attack. A similar observation concerning the generation of images from an adversarially trained classifier has been reported by Santurkar et al. (2019).

**The distance distribution** $p_{\text{dis}}$ relies on $\mathcal{D}(\mathbf{x}_{\text{ori}}, \mathbf{x}_{\text{adv}})$, representing the distance between $\mathbf{x}_{\text{adv}}$ and $\mathbf{x}_{\text{ori}}$. By its nature, samples that are closer to $\mathbf{x}_{\text{ori}}$ may yield a higher $p_{\text{dis}}$, which is consistent with the objective of generating adversarial samples. For example, if $\mathcal{D}$ represents the square of the $L_2$ norm, then $p_{\text{dis}}$ becomes a Gaussian distribution with a mean of $\mathbf{x}_{\text{ori}}$ and a variance determined by $c_1$. Figure 2(c) portrays samples drawn from $p_{\text{dis}}$ when $\mathcal{D}$ is the square of the $L_2$ distance. The samples closely resemble the original images, $\mathbf{x}_{\text{ori}}$s. This is attributed to the fact that each sample converges near the Gaussian distribution's mean, which corresponds to the $\mathbf{x}_{\text{ori}}$s.

**The prior distribution** $p_{\text{pri}}$ encapsulates our assumptions about the distribution of $\mathbf{x}_{\text{adv}}$. It is usually set as a constant, which corresponds to a uniform distribution, signifying a noninformative prior. In this paper, while we don't present an intricate choice for $p_{\text{pri}}$, we emphasize that it serves as a principle concept for incorporating prior knowledge about $\mathbf{x}_{\text{adv}}$ into the sampling process.

**The product of the distributions** Samples drawn from $p_{\text{adv}}$ tend to be concentrated in the regions of high density resulting from the product of $p_{\text{vic}}$ and $p_{\text{dis}}$, assuming $p_{\text{pri}}$ is constant. As is discussed, a robust victim classifier possesses generative capabilities. This means the high-density regions of $p_{\text{vic}}$ are inclined to generate images that embody the semantics of the target class. Conversely, the dense regions of $p_{\text{dis}}$ tend to produce images reflecting the semantics of the original image. If these high-density regions of both $p_{\text{vic}}$ and $p_{\text{dis}}$ intersect, then samples from $p_{\text{adv}}$ may encapsulate the semantics of both the target class and the original image. As depicted in Figure 3(a), the generated samples exhibit traces of both the target class and the original image. From our probabilistic perspective, the tendency of the generated adversarial samples to semantically resemble the target class stems from the generative ability of the victim distribution.

## 4 GENERATING SEMANTICS-AWARE ADVERSARIAL EXAMPLES

We propose semantic divergence, denoted by a non-symmetric divergence $\mathcal{D}_{\text{sem}}(\mathbf{x}_{\text{adv}}, \mathbf{x}_{\text{ori}}) := E(\mathbf{x}_{\text{adv}}; \mathbf{x}_{\text{ori}})$, where $E(\cdot; \mathbf{x}_{\text{ori}})$ represents the energy of an energy-based model trained on a dataset

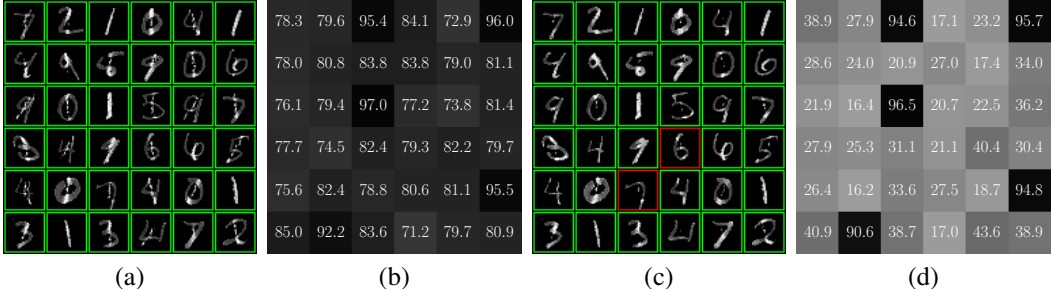

Figure 3: **(a)**: Samples from $p_{\text{adv}}(\cdot; \mathbf{x}_{\text{ori}}, y_{\text{tar}}) \propto \exp(-c_1 \cdot \mathcal{D}(\mathbf{x}_{\text{ori}}, \mathbf{x}_{\text{adv}})) \exp(-c_2 \cdot f(\mathbf{x}_{\text{adv}}, y_{\text{tar}}))$, where $\mathcal{D}$ is the $L_2$ norm, $f$ is the cross-entropy $f_{\text{CE}}$, $\mathbf{x}_{\text{ori}}$ are the first 36 images from the MNIST test set, $y_{\text{tar}}$ are set to 1, $c_1$ is $10^{-3}$, and $c_2$ is $10^{-2}$. **(c)**: Similar to (a), but with $f$ replaced by $f_{\text{CW}}$, as introduced in section 5.1. Essentially, this case applies the $L_2$ CW attack (Carlini & Wagner, 2017) using LMC instead of Adam optimization (We can call it prob CW). A green border indicates successful deception of the victim classifier, while a red border signifies failure. **(b) & (d)**: the predictive probability (softmax probability) of the target class, corresponding to each digit of Figures (a) and (c) on a one-to-one basis.

consisting of $\{t_1(\mathbf{x}_{\text{ori}}), t_2(\mathbf{x}_{\text{ori}}), \dots\}$. Here, $t_i \sim \mathcal{T}$, and $\mathcal{T}$ is a distribution of transformations that do not alter the original image's semantics. In practice, the choice of $\mathcal{T}$ depends on human subjectivity related to the dataset. Individuals are able to incorporate their personal comprehension of semantics into the model by designing their own $\mathcal{T}$. For instance, in the case of the MNIST dataset, the transformations could include scaling, rotation, distortion, and cropping, as illustrated in Figure 2(d). We assume that such transformations do not affect the semantics of the digits in the MNIST dataset. Consequently, our proposed semantic divergence induces the corresponding distance distribution $p_{\text{dis}}(\mathbf{x}_{\text{adv}}; \mathbf{x}_{\text{ori}}) \propto \exp(-c_1 \cdot E(\mathbf{x}_{\text{adv}}; \mathbf{x}_{\text{ori}}))$, which is exactly the distribution of the energy-based model introduced in formula (3). It is worth noting that the choice of $\mathcal{T}$ is subjective, for more details, see section 8 and appendix D.

By proposing semantic divergence, we successfully transformed simple distance distributions induced by geometric distances (such as the Gaussian distribution corresponding to the $L_2$ distance and the Laplace distribution for the $L_1$ distance) into trainable, data-driven distributions. This data-driven approach enables users to apply data augmentation or integrate supplementary data to characterize semantic distances, based on their personal interpretation of semantics. As a result, it transitions the geometric distance constraints in adversarial attacks to semantic-based constraints.

We claim that, given an appropriate $\mathcal{T}$, semantic divergence can surpass geometric distance. Empirically, when attempting to deceive a robust classifier, it's challenging to limit the geometric distance between the adversarial and original images without leaving traces of the adversarial attack, as depicted in Figure 1 and Figure 3. The attacked images either display a 'shadow' of the target digits or reveal conspicuous tampering traces, such as in Figure 3(c), where the attacked digit turns gray. This phenomenon was empirically observed and tested by Song et al. (2018) through an A/B test. Conversely, the samples from $p_{\text{adv}}$, as shown in Figure 4, scarcely display any evident signs of an adversarial attack. While semantic divergence can't entirely prevent the generation of a sample resembling the target class, as shown in Figure 4(a), we discuss certain techniques to mitigate this issue in Section 5.1.

A plausible interpretation is that, when using $L_1$ or $L_2$ distances, depending on a geometric distance-based distribution causes $p_{\text{dis}}(\cdot, \mathbf{x}_{\text{ori}})$ to cluster tightly around $\mathbf{x}_{\text{ori}}$. This results in a minimal overlap between $p_{\text{dis}}$ and $p_{\text{vic}}$, with the overlapping region located in the tail areas of the Gaussian or Laplace distribution. Consequently, image samples manifest a combination of Gaussian noise and features from the victim distribution. This mixture often gives rise to discernible "shadows" or evident markers of unnatural alterations. However, when using a distribution based on a proper semantic divergence, the overlap between $p_{\text{dis}}$ and $p_{\text{vic}}$ expands, occurring in regions that retain the desired semantics, while still effectively deceiving the classifier.

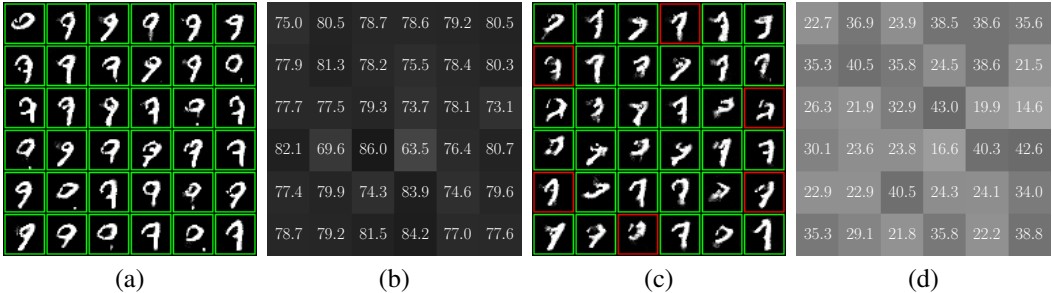

(a)  (b)  (c)  (d)

Figure 4: **(a) & (c)**: Samples from $p_{\text{adv}}(\cdot; \mathbf{x}_{\text{ori}}, y_{\text{tar}}) \propto \exp(-c_1 \cdot \mathcal{D}(\mathbf{x}_{\text{ori}}, \mathbf{x}_{\text{adv}})) \exp(-c_2 \cdot f(\mathbf{x}_{\text{adv}}, y_{\text{tar}}))$, where $\mathbf{x}_{\text{ori}}$ refers to the original image of digit "7" shown in Figure 1 and $y_{\text{tar}}$ refers to class 9. $\mathcal{D}$ represents our proposed semantic divergence. In (a), $f$ is the cross-entropy $f_{\text{CE}}$, while in (c), $f$ is $f_{\text{CW}}$. Constants are set as $c_1 = 1.0$ and $c_2 = 10^{-2}$. A green border indicates successful deception of the victim classifier, whereas a red border denotes failure. **(b) & (d)**: The predictive probability (softmax probability) of the target class, corresponding to each digit in Figures (a) and (c) on a one-to-one basis.

## 5 TECHNIQUES FOR DECEIVING ROBUST CLASSIFIERS

In this paper, we introduce four techniques that enhance the performance of our proposed method in generating high-quality adversarial examples. Due to space constraints, we detail only two of these techniques in this section. For the remaining two techniques, please refer to Appendix E.

### 5.1 VICTIM DISTRIBUTIONS

The victim distribution $p_{\text{vic}} \propto \exp(c_2 \cdot f(\mathbf{x}_{\text{adv}}, y_{\text{tar}}))$ is influenced by the choice of function $f$. Let $g_\phi : [0,1]^n \to \mathbb{R}^{|\mathcal{Y}|}$ be a classifier that produces logits as output with $\phi$ representing the neural network parameters, $n$ denoting the dimensions of the input, and $\mathcal{Y}$ being the set of labels (the output of $g_\phi$ are logits). Szegedy et al. (2013) suggested using cross-entropy as the function $f$, which can be expressed as

$$f_{\text{CE}}(\mathbf{x}, y_{\text{tar}}) := -g_\phi(\mathbf{x})[y_{\text{tar}}] + \log \sum_y \exp(g_\phi(\mathbf{x})[y]) = -\log \sigma(g_\phi(\mathbf{x}))[y_{\text{tar}}]$$

where $\sigma$ denotes the softmax function.

Carlini & Wagner (2017) explored and compared multiple options for $f$. They found that, empirically, the most efficient choice of their proposed $f$s is:

$$f_{\text{CW}}(\mathbf{x}, y_{\text{tar}}) := \max(\max_{y \neq y_{\text{tar}}} g_\phi(\mathbf{x})[y] - g_\phi(\mathbf{x})[y_{\text{tar}}], 0).$$

From Figure 3 and Figure 4, we observe that $f_{\text{CW}}$ outperforms $f_{\text{CE}}$ when the $p_{\text{dis}}$ depends on either geometric distance or semantic divergence. A potential explanation for this phenomenon is that, according to its definition, $f_{\text{CW}}$ becomes 0 if the classifier is successfully deceived during the iteration process. This setting ensures that the generator does not strive for a relatively high softmax probability for the target class; it simply needs to reach a point where the victim classifier perceives the image as belonging to the target class. Consequently, after the iteration, the victim classifier assigns a relatively low predictive probability to the target class $\sigma(g_\phi(\mathbf{x}_{\text{adv}}))[y_{\text{tar}}]$, as demonstrated in Figure 3(d) and Figure 4(d).

In this study, we introduce two additional choices for the function $f$. Although these alternatives are not as effective as $f_{\text{CW}}$, we present them in Appendix I for further exploration.

### 5.2 DATA AUGMENTATION BY THIN PLATE SPLINES (TPS) DEFORMATION

Thin-plate-spline (TPS) (Bookstein, 1989) is a commonly used image deforming method. Given a pair of control points and target points, TPS computes a smooth transformation that maps the control points to the target points, minimizing the bending energy of the transformation. This process results

in localized deformations while preserving the overall structure of the image, making TPS a valuable tool for data augmentation.

As introduced in Section 4, we aim to train an energy-based model on transformations of a single image $\mathbf{x}_{\text{ori}}$. In practice, if the diversity of the augmentations of $\mathbf{x}_{\text{ori}}$, represented as $t(\mathbf{x}_{\text{ori}})$, is insufficient, the training of the probabilistic generative model is prone to overfitting. To address this issue, we use TPS as a data augmentation method to increase the diversity of $t(\mathbf{x}_{\text{ori}})$. For each $\mathbf{x}_{\text{ori}}$, we set a $5 \times 5$ grid of source control points, $\mathcal{P}_{\text{sou}} = \{(x^{(i)}, y^{(i)})\}_{i=1}^{5 \times 5}$, and defining the target points as $\mathcal{P}_{\text{tar}} = \{(x^{(i)} + \epsilon_x^{(i)}, y^{(i)} + \epsilon_y^{(i)})\}_{i=1}^{5 \times 5}$, where $\epsilon_x^{(i)}, \epsilon_y^{(i)} \sim \mathcal{N}(0, \sigma^2)$ are random noise added to the source control points. We then apply TPS transformation to $\mathbf{x}_{\text{ori}}$ with $\mathcal{P}_{\text{sou}}$ and $\mathcal{P}_{\text{tar}}$ as its parameters. This procedure is depicted in Figure 5. By setting an appropriate $\sigma$, we can substantially increase the diversity of the one-image dataset while maintaining its semantic content.

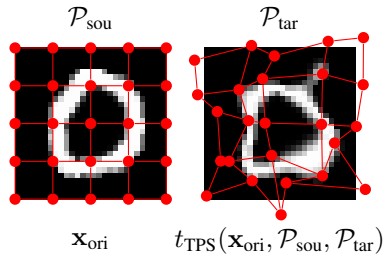

$\mathcal{P}_{\text{sou}}$  $\mathcal{P}_{\text{tar}}$

$\mathbf{x}_{\text{ori}}$  $t_{\text{TPS}}(\mathbf{x}_{\text{ori}}, \mathcal{P}_{\text{sou}}, \mathcal{P}_{\text{tar}})$

Figure 5: TPS as a data augmentation. **Left**: The original image $\mathbf{x}_{\text{ori}}$ superimposed with a $5 \times 5$ grid of source control points $\mathcal{P}_{\text{sou}}$. **Right**: The transformed image overlaid with a grid of target control points $\mathcal{P}_{\text{tar}}$.

# 6 EXPERIMENT

## 6.1 IMPLEMENTATION

We implemented our proposed semantics-aware adversarial attack on three datasets: MNIST, SVHN and CIFAR10. For MNIST, the victim classifier we used was an adversarially trained MadryNet (Madry et al., 2017). For SVHN and CIFAR10, we utilized an adversarially trained ResNet18, in accordance with the methodology outlined by Song et al. (2018). On the distance distribution side, for every original image $\mathbf{x}_{\text{ori}}$, we trained an energy-based model on the training set, which is represented as $\{t_1(\mathbf{x}_{\text{ori}}), t_2(\mathbf{x}_{\text{ori}}), \dots\}$. In this case, $t_i$ follows a distribution of transformations, $\mathcal{T}$, that do not change the semantics of $\mathbf{x}_{\text{ori}}$. For MNIST, we characterized $\mathcal{T}_{\text{MNIST}}$ as including Thin Plate Spline (TPS) transformations, scaling, and rotation. For SVHN, we defined $\mathcal{T}_{\text{SVHN}}$ as comprising TPS transformations and alterations in brightness and hue. For CIFAR10, $\mathcal{T}_{\text{CIFAR10}}$ contains TPS only. Detailed specifics related to our implementation can be found in Appendix G.

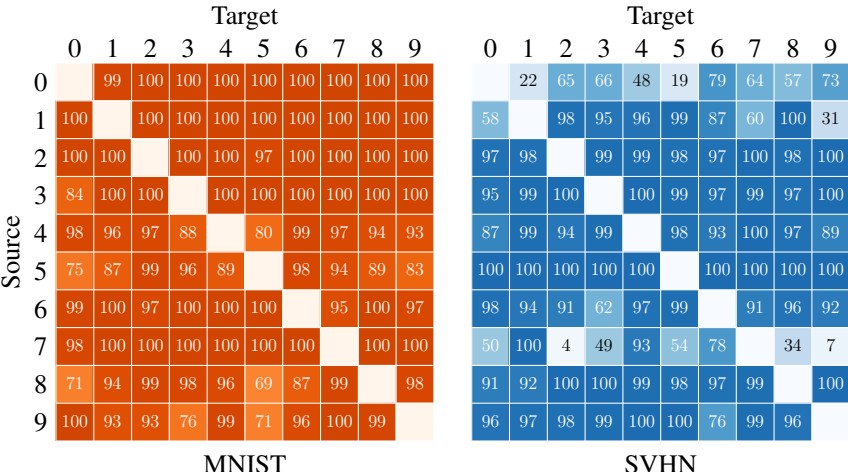

Figure 6: The success rates (%) of our targeted adversarial attack. Corresponding sample examples for each grid are depicted in the top right and bottom right sections of Figure 1. Refer to Table 1 for overall success rate.

Table 1: Success rate comparison between the method proposed by Song et al. (2018) and ours. The results presented in this table are for reference only, as Song's results are taken directly from their paper, and we did not use the same group of annotators for our evaluation.

| Robust Classifier | Success Rate of Song et al. (2018) | Our Success Rate |
|---|---|---|
| MadryNet (Madry et al., 2017) on MNIST | 85.2 | **96.2** |
| ResNet18 (He et al., 2016) (adv-trained) on SVHN | 84.2 | **86.3** |

## 6.2 Targeted attacks against adversarial training

**MNIST / SVHN**  Our method generates adversarial samples that can deceive classifiers, but it does not guarantee the preservation of the original image's semantic meaning. As such, we consider an adversarial example successful if human annotators perceive it as having the same meaning as the original label, in line with the approach by Song et al. (2018). In detail, we randomly select 10 digits, each representing a different class, from the MNIST/SVHN test set to serve as the original image $\mathbf{x}_{ori}$. These are depicted on the left side of Figure 1. For each $\mathbf{x}_{ori}$, we iterate through the target class $y_{tar}$ ranging from 0 to 9, excluding the class $y_{ori}$ that signifies the ground-truth label of $\mathbf{x}_{ori}$. For every pair of $\mathbf{x}_{ori}$ and $y_{tar}$, we generate $N = 100$ adversarial examples post sample refinement. The result of each pair is illustrated in Figure 6. The overall success rate is illustrated in Table 1. The detail of calculating the success rate is in Appendix F.

**CIFAR10**  As illustrated in Figure 1, for CIFAR10, both PGD and our approach generate adversarial examples that largely maintain the original image's semantics. To emphasize the superiority of our method, we undertake a human-driven comparison, juxtaposing images from PGD-$L_2$ and our method, as well as PGD-$L_\infty$ and our method. In detail, by selecting an appropriate value for $\epsilon$, we ensure that in most cases, both PGD attack variants can deceive the classifier, as demonstrated in Figure 1. Under these circumstances, we extract examples that successfully mislead the classifier. We then pair images produced by the PGD attack with those generated by our method and present them to 10 annotators. They are tasked with determining which image appears more natural (i.e., without signs of computational tampering). The interface used by the annotators can be found in Appendix H. When comparing our method to PGD-$L_\infty$, annotators found that $96.3\%$ of samples produced by our technique were more similar to real images. Likewise, in the comparison between our method and PGD-$L_2$, $87.4\%$ of the time, annotators felt that our method's samples were more akin to real images.

## 6.3 Transferability

Table 2 and Table 3 shows the transferability of our proposed method. We attack adversarially trained Madry Net with our method and feed legitimate unrestricted adversarial examples, as verified by annotators, to other classifiers.

Table 2: Transferability of our proposed method on MNIST. Numbers represent accuracies of classifiers.

| Classifier
Attack Type | Madry Net (no adv) | Madry Net (adv) | ResNet (no adv) | ResNet (adv) | Certified defence Wong & Kolter (2018) |
|---|---|---|---|---|---|
| No attack | 99.5 | 98.4 | 99.3 | 99.4 | 98.2 |
| Song et al. (2018) (w/o noise) | 95.1 | 0 | 92.7 | 93.7 | 84.3 |
| Song et al. (2018) (w/ noise) | 78.3 | 0 | **73.8** | 84.9 | 63.0 |
| Our method | **38.3** | 0 | 76.8 | **82.5** | **60.5** |

## 7 Related work

**Unrestricted adversarial examples**  Song et al. (2018) proposed generating unrestricted adversarial examples from scratch using conditional generative models. In their work, the term "unrestricted"

Table 3: Transferability of our proposed method on SVHN and CIFAR10.

| Dataset | Attack Type | ResNet18 (no adv) | ResNet18 (adv) | VGG19 (no adv) | VGG19 (adv) | Certified defence Wong & Kolter (2018) |
|---|---|---|---|---|---|---|
| SVHN | No attack | 95.2 | 93.6 | 94.0 | 95.0 | 79.6 |
| | Our method | **26.0** | 0 | **32.0** | **46.9** | **40.4** |
| CIFAR10 | No attack | 93.0 | 83.5 | 92.6 | 81.0 | 73.0 |
| | Our method | **33.4** | 0 | **47.5** | **55.7** | **42.1** |

indicates that the generated adversarial samples, $\mathbf{x}_{\text{adv}}$, are not restricted by a geometric distance such as the $L_2$ norm or $L_\infty$ norm. The key difference between their approach and ours is that their adversarial examples $\mathbf{x}_{\text{adv}}$ are independent of any specific $\mathbf{x}_{\text{ori}}$, while our model generates $\mathbf{x}_{\text{adv}}$ based on a given $\mathbf{x}_{\text{ori}}$. By slightly modifying (4), we can easily incorporate Song's "unrestricted adversarial examples" into our probabilistic perspective:

$$p_{\text{adv}}(\mathbf{x}_{\text{adv}}; y_{\text{sou}}, y_{\text{tar}}) \propto p_{\text{vic}}(\mathbf{x}_{\text{adv}}; y_{\text{tar}})p_{\text{dis}}(\mathbf{x}_{\text{adv}}; y_{\text{sou}})p_{\text{pri}}(\mathbf{x}_{\text{adv}}) \qquad (5)$$

where $y_{\text{sou}}$ is the source class. It becomes evident that the adversarial examples generated by our $p_{\text{adv}}(\cdot; \mathbf{x}_{\text{ori}}, y_{\text{tar}})$ adhere to Song's definition when $\mathbf{x}_{\text{ori}}$ is labeled as $y_{\text{sou}}$. By training class-conditional energy-based model we are able to generate adversarial samples from this $p_{\text{adv}}$, see Appendix L. Discussions of other works on unrestricted adversarial examples (Xiao et al., 2018; Bhattad et al., 2019; Joshi et al., 2019; Hosseini & Poovendran, 2018) can be found in Appendix M.

**TPS as a Data Augmentation Technique**    To the best of our knowledge, Vinker et al. (2021) were the first to employ TPS as a data augmentation method. They utilized TPS as a data augmentation strategy in their generative model for conditional image manipulation based on a single image.

## 8 Discussion

**MNIST and SVHN are not 'easy'**    At first glance, MNIST and SVHN might appear simpler due to their low resolution and basic structure, especially when compared to natural images. Contrary to this initial impression, they pose a greater challenge than natural images. This is because the digit data lacks diversity. Consequently, robust classifiers can easily memorize the distinct shape of each digit, making them particularly resistant to attacks, as illustrated in Figure 2(b). As depicted in Figure 1, more significant transformations are required to successfully deceive the classifiers on MNIST and SVHN.

**Limitations of Energy-Based Models**    A limitation of this work is the challenges associated with training energy-based models (EBMs). As highlighted by earlier research from Du & Mordatch (2019) and Grathwohl et al. (2019), the process of training EBMs can be intricate and demanding. There remains a discernible difference in the generation quality between EBMs and other popular probabilistic generative models, such as variational autoencoders and diffusion models. As a result, our current framework struggles to produce adversarial samples for higher-resolution images. However, given the ongoing advancements in deep learning and generative models, we are optimistic that this challenge will be addressed in the near future.

## 9 Conclusion

In this work, we present a novel probabilistic perspective on adversarial examples. Building on this probabilistic perspective, we introduce semantic divergence as an alternative to the commonly used geometric distance. We also propose corresponding techniques for generating semantically-aware adversarial examples. Human participation experiments indicate that our proposed method can often deceive robust classifiers while maintaining the original semantics of the input, although not in all cases. We believe that our new perspective and methodology will pave the way for a fresh paradigm in adversarial attacks.

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

# A PRELIMINARIES (CONTINUED)

## A.1 LANGEVIN MONTE CARLO (LMC)

Langevin Monte Carlo (also known as Langevin dynamics) is an iterative method that could be used to find near-minimal points of a non-convex function $g$ (Raginsky et al., 2017; Zhang et al., 2017; Tzen et al., 2018; Roberts & Tweedie, 1996). It involves updating the function as follows:

$$\mathbf{x}_0 \sim p_0, \quad \mathbf{x}_{t+1} = \mathbf{x}_t - \frac{\epsilon^2}{2}\nabla_x g(\mathbf{x}_t) + \epsilon\mathbf{z}_t, \quad \mathbf{z}_t \sim \mathcal{N}(0, I) \tag{6}$$

where $p_0$ could be a uniform distribution. Under certain conditions on the drift coefficient $\nabla_x g$, it has been demonstrated that the distribution of $\mathbf{x}_t$ in (6) converges to its stationary distribution (Chiang et al., 1987; Roberts & Tweedie, 1996), also referred to as the Gibbs distribution $p(\mathbf{x}) \propto \exp(-g(\mathbf{x}))$. This distribution concentrates around the global minimum of $g$(Gelfand & Mitter, 1991; Xu et al., 2018; Roberts & Tweedie, 1996). If we choose $g$ to be $E_\theta$, then the stationary distribution corresponds exactly to the EBM's distribution defined in (3). As a result, we can draw samples from the EBM using LMC. By replacing the exact gradient with a stochastic gradient, we obtain Stochastic Gradient Langevin Dynamics (SGLD) (Welling & Teh, 2011; Teh et al., 2016).

## A.2 TRAINING EBM

To train an EBM, we aim to minimize the minus expectation of the log-likelihood, represented by

$$\mathcal{L}_{\text{EBM}} = \mathbb{E}_{X\sim p_d}[-\log p_\theta(X)] = \mathbb{E}_{X\sim p_d}[E_\theta(X)] - \log Z_\theta$$

where $p_d$ is the data distribution. The gradient is

$$\nabla_\theta \mathcal{L}_{\text{EBM}} = \mathbb{E}_{X\sim p_d}[\nabla_\theta E_\theta(X)] - \nabla_\theta \log Z_\theta = \mathbb{E}_{X\sim p_d}[\nabla_\theta E_\theta(X)] - \mathbb{E}_{X\sim p_\theta}[\nabla_\theta E_\theta(X)] \tag{7}$$

(see (Song & Kingma, 2021) for derivation). The first term of $\nabla_\theta \mathcal{L}_{\text{EBM}}$ can be easily calculated as $p_d$ is the distribution of the training set. For the second term, we can use LMC to sample from $p_\theta$ (Hinton, 2002).

Effective training of an energy-based model (EBM) typically requires the use of techniques such as sample buffering and regularization. For more information, refer to the work of Du & Mordatch (2019).

# B PROOF OF THE THEOREM

**Theorem.** *Given the condition that $p_{vic}(\mathbf{x}_{adv}; y_{tar}) \propto \exp(-c_2 \cdot f(\mathbf{x}_{adv}, y_{tar}))$, $p_{dis}(\mathbf{x}_{adv}; \mathbf{x}_{ori}) \propto \exp(-c_1 \cdot \mathcal{D}(\mathbf{x}_{ori}, \mathbf{x}_{adv}))$, and set $p_{pri}(\mathbf{x}_{adv})$ as a constant, the samples drawn from $p_{adv}$ will exhibit the same distribution as the adversarial examples derived from applying the box-constrained Langevin Monte Carlo method to the optimization problem delineated in equation (1).*

*Proof.* Lamperski (2021) introduced the Projected Stochastic Gradient Langevin Algorithms (PS-GLA) to address box-constraint optimization problems. By leveraging the PSGLA, we can generate samples close to the solution of the optimization problem as stated in Equation (1). This leads us to the following update rule:

$$\mathbf{x}_0 \sim p_0, \quad \mathbf{x}_{t+1} = \Pi_{[0,1]^n}\left(\mathbf{x}_t - \frac{\epsilon^2}{2}\nabla_x\mathcal{L}(\mathbf{x}_t, y_{\text{tar}}) + \epsilon\mathbf{z}_t\right), \quad \mathbf{z}_t \sim \mathcal{N}(0, I) \tag{8}$$

where $\Pi[0, 1]^n$ is a projection that clamps values within the interval $[0, 1]^n$. According to Lamperski (2021), samples generated via this update rule will converge to a stationary distribution, which can be termed the Gibbs distribution $p_{\text{gibbs}}$:

$$\begin{aligned}
p_{\text{gibbs}}(\mathbf{x}_{\text{adv}}; y_{\text{tar}}) &\propto \exp(-\mathcal{L}(\mathbf{x}_{\text{adv}}, y_{\text{tar}})) \\
&\propto \exp(-c_1 \cdot \mathcal{D}(\mathbf{x}_{\text{ori}}, \mathbf{x}_{\text{adv}}) - c_2 \cdot f(\mathbf{x}_{\text{adv}}, y_{\text{tar}})) \cdot 1 \\
&\propto \exp(-c_1 \cdot \mathcal{D}(\mathbf{x}_{\text{ori}}, \mathbf{x}_{\text{adv}})) \cdot \exp(-c_2 \cdot f(\mathbf{x}_{\text{adv}}, y_{\text{tar}})) \cdot 1 \\
&\propto p_{\text{dis}}(\mathbf{x}_{\text{adv}}; \mathbf{x}_{\text{ori}})p_{\text{vic}}(\mathbf{x}_{\text{adv}}; y_{\text{tar}})p_{\text{pri}}(\mathbf{x}_{\text{adv}})
\end{aligned}$$

which matches the form of $p_{\text{adv}}$. It is a well-established fact that random variables with identical unnormalized probability density functions share the same distribution. □

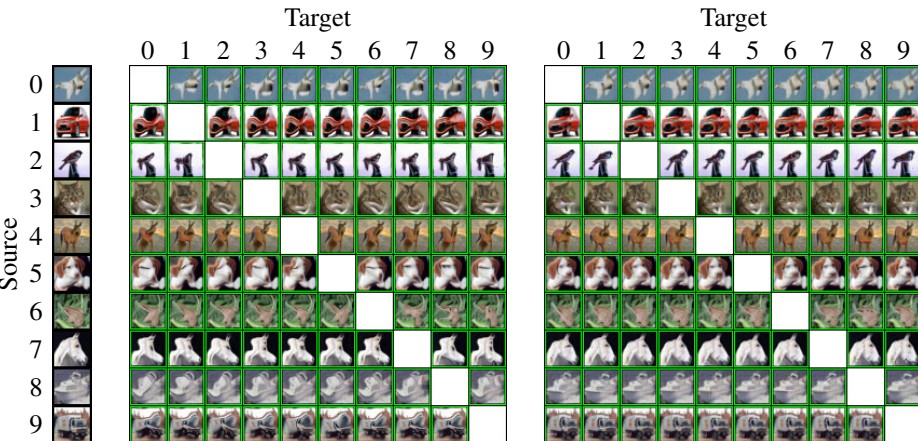

Figure 7: Adversarial examples generated by our proposed method with varying TPS variance. Left: TPS standard deviation set at $0.12$; Right: TPS standard deviation set at $0.05$.

## C   SAMPLE FROM ADVERSARIAL DISTRIBUTIONS

In this paper, the distributions we often refer to (e.g., $p_{\text{adv}}$, $p_{\text{vic}}$, $p_{\text{dis}}$, and $p_{\text{pri}}$) typically take the unnormalized form $\exp(-c \cdot g(x))$ or a constant. Sampling from the product of these distributions can be efficiently done using Langevin Monte Carlo, specifically its box-constraint variant, PSGLA. For instance, consider sampling from a distribution $p(\mathbf{x}) \propto \exp(-c_1 g_1(\mathbf{x})) \exp(-c_2 g_2(\mathbf{x}))$. This can be reformulated as $p(\mathbf{x}) \propto \exp(-c_1 g_1(\mathbf{x}) - c_2 g_2(\mathbf{x}))$. Notably, this distribution mirrors the Gibbs distribution that results from applying PSGLA to the potential $\mathcal{L} = -c_1 g_1(\mathbf{x}) - c_2 g_2(\mathbf{x})$. Thus, samples can be obtained using the update rule in equation (8) with $\mathcal{L} = -c_1 g_1(\mathbf{x}) - c_2 g_2(\mathbf{x})$.

## D   CHOICE OF THE SUBJECTIVE TRANSFORMATIONS

In Section 4, we mentioned that the distribution of transformations, $\mathcal{T}$, is influenced by human subjectivity. It's worth noting that what one individual perceives as a non-semantically-altering transformation might not be acceptable to another. For instance, as illustrated in Figure 7, one person might believe that the transformation on the left doesn't hinder their ability to discern the object's semantics and finds this degree of TPS transformation acceptable. However, another individual might perceive the left transformation as overly distorted, and only the more subtle TPS transformation on the right preserves the unchanged semantics. Thus, in practice, it's essential to select a transformation that not only maintains semantics but is also widely accepted by the general populace.

## E   TECHNIQUES FOR DECEIVING ROBUST CLASSIFIERS (CONTINUE)

### E.1   REJECTION SAMPLING

Directly sampling from $p_{\text{adv}}(\cdot; \mathbf{x}_{\text{ori}}, y_{\text{tar}})$ does not guarantee the generation of samples capable of effectively deceiving the classifier. To overcome this issue, we adopt rejection sampling (Von Neumann, 1951), which eliminates unsuccessful samples and ultimately yields samples from $p_{\text{adv}}(\mathbf{x}_{\text{adv}} | \arg\max_y g_\phi(\mathbf{x}_{\text{adv}})[y] = y_{\text{tar}}; \mathbf{x}_{\text{ori}}, y_{\text{tar}})$.

### E.2   SAMPLE REFINEMENT

After rejection sampling, the samples are confirmed to successfully deceive the classifier. However, not all of them possess high visual quality, as demonstrated in Figure 4(c). To automatically obtain $N$ semantically valid samples[1], we first generate $M$ samples from the adversarial distribution. Following rejection sampling, we sort the remaining samples and select the top $\kappa$ percent based on the softmax

---

[1]In practice, we could select adversarial samples by hand, but we focus on automatic selection here.

probability of the original image's class, as determined by an auxiliary classifier. Finally, we choose the top $N$ samples with the lowest energy $E$, meaning they have the highest likelihood according to the energy-based model.

The auxiliary classifier is trained on the data-augmented training set. We do not use the energy of the samples as the sole criterion for selection because some low-visual quality samples may also have a high likelihood. This occurrence is further explained and examined in Appendix J. The entire process of rejection sampling and sample refinement is portrayed in Algorithm 1.

---

**Algorithm 1** Rejection Sampling and Sample Refinement

---

**Input:** A trained energy based model $E(\cdot; \mathbf{x}_{\text{ori}})$ based on the original image $\mathbf{x}_{\text{ori}}$, the victim classifier $g_\phi$, an auxiliary classifier $g_\psi$, number of initial samples $M$, number of final samples $N$, the percentage $\kappa$.

**Output:** $N$ adversarial samples $\mathbf{x}$.

  $\mathbf{x} = \emptyset$
  **for** $0 \le i < M$ **do**
    $\mathbf{x}_{\text{adv}} \sim p_{\text{adv}}(\cdot; \mathbf{x}_{\text{ori}}, y_{\text{tar}})$             $\triangleright$ Sample from the adversarial distribution.
    **if** $\arg\max_y g_\phi(\mathbf{x}_{\text{adv}})[y] = y_{\text{tar}}$ **then**     $\triangleright$ Accept if $\mathbf{x}_{\text{adv}}$ deceive the classifier.
      $\mathbf{x} = \mathbf{x} \cup \{\mathbf{x}_{\text{adv}}\}$
    **end if**
  **end for**
  Sort $\mathbf{x}$ by $\sigma(g_\psi(\mathbf{x}_i))[y_{\text{ori}}]$ for $i \in \{1, \dots, |\mathbf{x}|\}$ in descent order
  $\mathbf{x} = (\mathbf{x}_i)_{i=1}^{\lfloor \kappa|\mathbf{x}| \rfloor}$            $\triangleright$ Select the first $\kappa$ percent elements from $\mathbf{x}$.
  Sort $\mathbf{x}$ by $E(\mathbf{x}_i; \mathbf{x}_{\text{ori}})$ for $i \in \{1, \dots, |\mathbf{x}|\}$ in ascent order
  $\mathbf{x} = (\mathbf{x}_i)_{i=1}^{N}$               $\triangleright$ Select the first $N$ elements from $\mathbf{x}$.

---

## F   CALCULATING THE SUCCESS RATE

To enhance the signal-to-noise ratio, we assign the same image to five different annotators and use the majority vote as the human decision, as done in (Song et al., 2018). The screenshot of the annotator's interface is in Appendix H.

In detail, we begin with an original image $\mathbf{x}_{\text{ori}}$, its label $y_{\text{ori}}$, and a target class $y_{\text{tar}}$. We draw $M = 2000$ samples from $p_{\text{adv}}(\cdot; \mathbf{x}_{\text{ori}}, y_{\text{tar}})$, rejecting those that fail to deceive the victim classifier. After sample refinement, we obtain $N = 100$ adversarial examples, $\mathbf{x}_{\text{adv}}^{(i)}$ for $i \in \{1, \dots, N\}$. We express the human annotators' decision as function $h$ and derive the human decision $y_{\text{hum}}^{(i)} = h(\mathbf{x}_{\text{adv}}^{(i)})$. As previously mentioned, an adversarial example $\mathbf{x}_{\text{adv}}^{(i)}$ is considered successful if $y_{\text{hum}}^{(i)}$ is equal to $y_{\text{ori}}$. We then compute the success rate $s$ as follows:

$$s = \frac{\sum_{i=1}^{N} \mathbb{1}(y_{\text{hum}}^{(i)} = y_{\text{ori}})}{N}$$

where $\mathbb{1}$ represents the indicator function.

## G   IMPLEMENTATION DETAILS

In this section, we delve into the specifics of this study's implementation. The related source code has been provided within the supplementary materials. In case of any remaining uncertainties, please refer to the included source code for further clarity.

### G.1   VICTIM CLASSIFIERS

We utilize two distinct classifiers as the victim classifiers, each with a specific dataset: MadryNet (Madry et al., 2017) with MNIST, and ResNet18 (He et al., 2016) with the SVHN and CIFAR10 dataset.

### G.1.1 MADRYNET

We utilize MadryNet (Madry et al., 2017), a convolutional neural network (CNN), as the victim classifier for the MNIST dataset. The MadryNet model is trained using the Adam optimizer, with a learning rate set at $10^{-4}$. Our training regimen comprises 14 epochs, with a batch size of 64. Any additional hyperparameters are retained at their default settings as prescribed by PyTorch.

During adversarial training on MadryNet, we implement a Projected Gradient Descent (PGD) untargeted attack on the training data, using the parameters $\epsilon = 0.3, \alpha = 0.036$, and 10 steps.

### G.1.2 RESNET18

In the case of the SVHN and CIFAR10 dataset, we employ ResNet18 (He et al., 2016), another well-known architecture. The ResNet18 model is trained using the Adam optimizer, with a learning rate set at $10^{-4}$. Our training regimen comprises 14 epochs, with a batch size of 64. Any additional hyperparameters are retained at their default settings as prescribed by PyTorch. Similarly to the MadryNet, adversarial training on ResNet18 also involves a PGD untargeted attack on the training data, but with different parameters: $\epsilon = 0.03, \alpha = 0.01$, and 10 steps.

## G.2 ENERGY-BASED MODELS

### G.2.1 NEURAL NETWORK STRUCTURE

For the MNIST dataset, we utilize a specialized convolutional neural network with the undermentioned architecture:

- The model commences with a 2D convolutional layer employing 64 filters of 5x5 kernel size, a stride of 2, and a larger padding of 4, effectively augmenting the input image size to 32x32. A 'Swish' activation function is then invoked to incorporate non-linearity.

- The second layer consists of another convolutional layer using 128 filters of 3x3 size, with a stride of 2 and padding of 1, followed by the 'Swish' activation function.

- The third layer is a replica of the previous one but escalates the filter count to 256 while preserving the filter size, stride, and padding, followed by a 'Swish' activation.

- The fourth convolutional layer utilizes 256 filters, similar to the third layer, with a 3x3 kernel size, stride of 2, and padding of 1. This is succeeded by a 'Swish' activation function.

- Post convolution, the output undergoes flattening to eliminate spatial dimensions.

- The flattened output is then passed through a fully connected layer with 256 units, followed by the 'Swish' activation function.

- The architecture culminates with a second fully connected layer mapping the 256 units to a determined output size. This size usually correlates to the number of classes in a classification task or the desired output size in regression tasks.

The neural network training employs the Adam optimizer with a learning rate of $10^{-4}$, batch size of 128, and 200 epochs.

For the SVHN and CIFAR10 dataset, we utilize the WideResNet structure, specifically, the WRN-28-10 variant. Training the WRN-28-10 also involves the Adam optimizer with a learning rate of $10^{-4}$, batch size of 128, and spans across 50 epochs.

## G.3 HARDWARE SPECIFICATIONS

All of our experiments were conducted on a machine equipped with an Intel E5-2680 v3 CPU and an NVIDIA RTX 3090 GPU.

# H ANNOTATOR INTERFACE

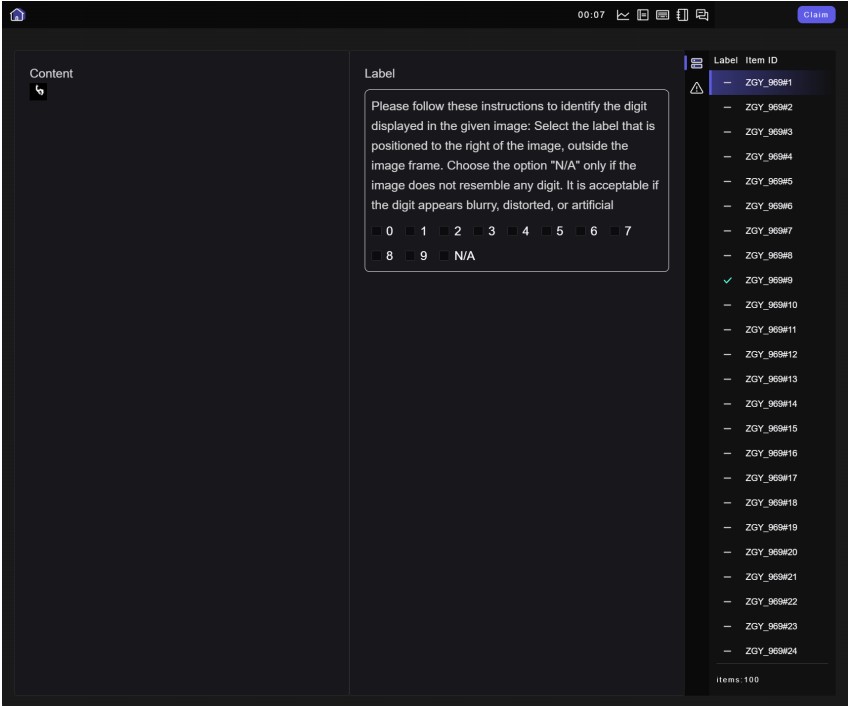

Figure 8: Annotator Interface for image annotation

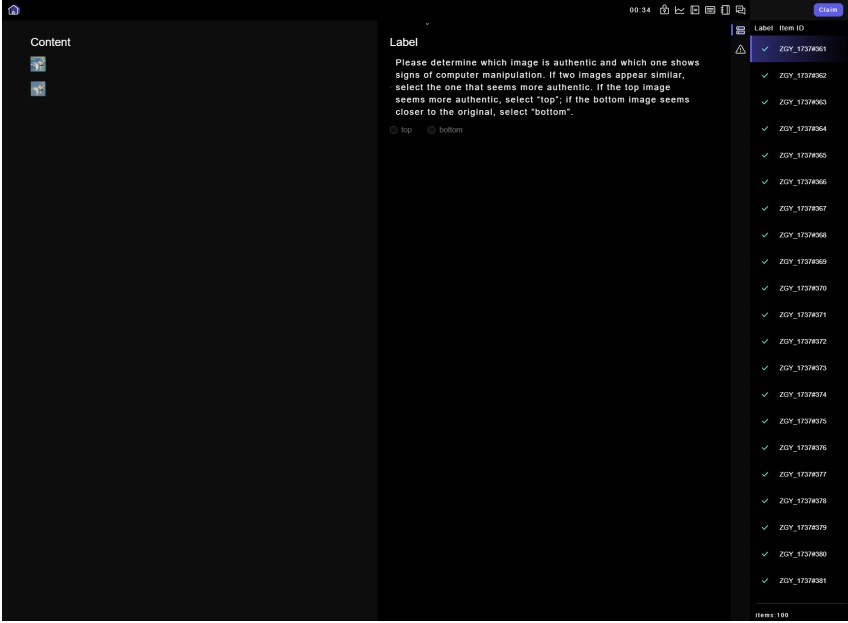

Figure 9: Annotator Interface for comparison

# I  CHOOSING A PROPER F

In this section, we propose two additional $f$ functions, where the first is based on predictive entropy and the second is rooted in joint-energy.

The Predictive Entropy based $f$, denoted as $f_{\text{PE}}$, is formulated as follows:

$$f_{\text{PE}}(\mathbf{x}, y_{\text{tar}}) := -c_{\text{PE}} \sum_y \sigma(g_\phi(\mathbf{x}))[y] \log \sigma(g_\phi(\mathbf{x}))[y] + f_{\text{CE}}(\mathbf{x}, y_{\text{tar}})$$

Here, $c_{\text{PE}}$ is a constant that determines the weight of the predictive entropy.

On the other hand, the Joint-Energy based $f$, denoted as $f_{\text{JE}}$, is given by:

$$f_{\text{JE}}(\mathbf{x}, y_{\text{tar}}) := -g_\phi(\mathbf{x})[y_{\text{tar}}] + c_{\text{JE}} \log \sum_y \exp(g_\phi(\mathbf{x})[y])$$

In this case, $c_{\text{JE}}$ is a constant controlling the weight of the logsumexp term. It is worth noting that when $c_{\text{JE}} = 1$, $f_{\text{JE}}$ simplifies to $f_{\text{CE}}$.

As shown in Figure 10, when $p_{\text{dis}}$ is fixed, choosing $f_{\text{CW}}$ results in better generation compared to $f_{\text{CE}}$, $f_{\text{PE}}$, and $f_{\text{JE}}$.

An interesting visual interpretation of this phenomenon can be found in Figure 11. Here, we draw samples from $p_{\text{vic}}(\cdot; y_{\text{tar}})$, observing that the samples drawn from the $p_{\text{vic}}$ induced by $f_{\text{CW}}$ contain the least semantic information.

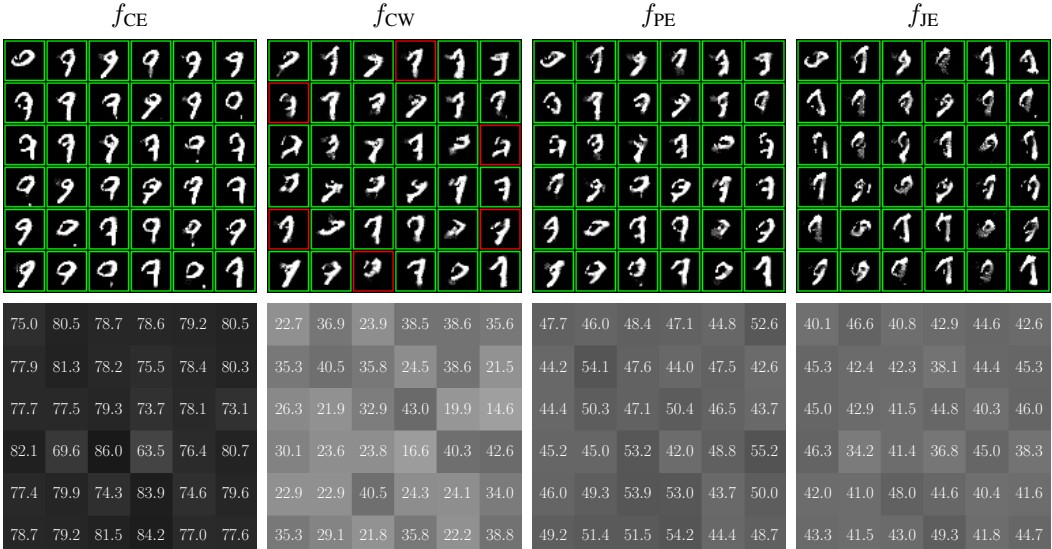

Figure 10: Targeted Attack on a Single Image: The source image belongs to class 7, and the target class is 9. The first row displays samples drawn from $p_{\text{adv}}(\cdot, \mathbf{x}_{\text{ori}}, y_{\text{tar}})$, where $\mathbf{x}_{\text{ori}}$ is an image from class 7 and $y_{\text{tar}} = 9$. All cases share the same random seed and the same $p_{\text{dis}}(\cdot; \mathbf{x}_{\text{ori}})$ trained on augmentations of $\mathbf{x}_{\text{ori}}$. The key distinction among the plots is the function $f$ used, in this order: $f_{\text{CE}}$, $f_{\text{CW}}$, $f_{\text{PE}}$, $f_{\text{JE}}$. The second row of plots showcases the predictive probability (softmax probability) of the target class, corresponding to each digit in the first row on a one-to-one basis. A green border signifies a successful deception of the victim classifier, while a red border indicates failure.

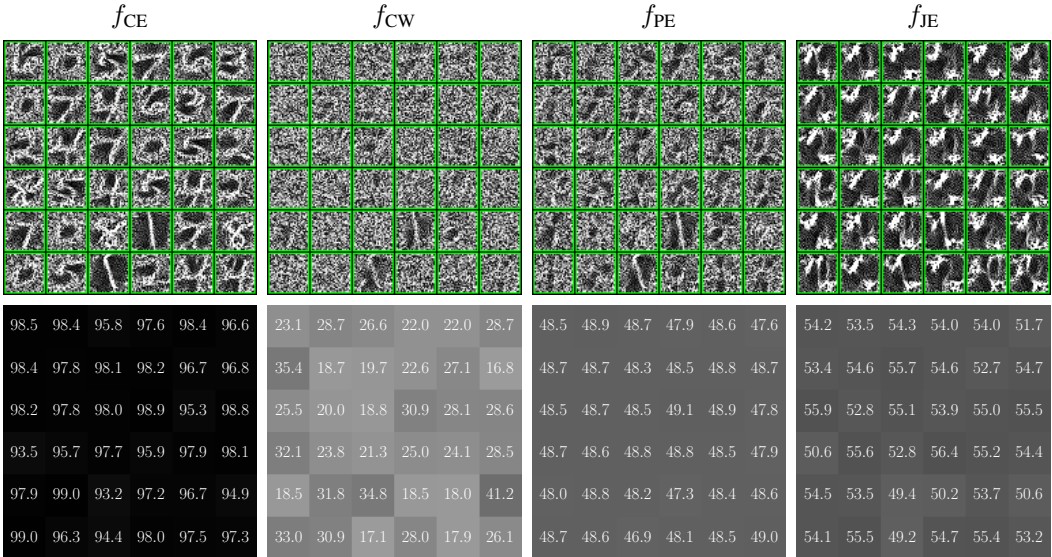

Figure 11: Samples drawn from the victim distribution $p_{vic}(\cdot; y_{tar})$ with randomly sampled $y_{tar}$. All four cases share the same random seed. The parameters are consistent with those in Figure 10.

## J LOW-VISUAL QUALITY SAMPLES WITH HIGH LIKELIHOOD

During our experiments, we observed that high likelihood samples do not invariably exhibit high visual quality. This phenomenon is showcased in Figure 12, where, despite being sorted by energy (a parameter proportional to likelihood), the earliest samples do not always deliver high visual quality. Further, we empirically identified a pattern: high likelihood samples that possess low visual quality often correspond to a low softmax probability for class $y_{ori}$ (the label of the original image). Leveraging this observation, we decided to retain only the top few percentile of samples that have the highest softmax probability for class $y_{ori}$ within an auxiliary classifier, and then sort the remaining samples by energy.

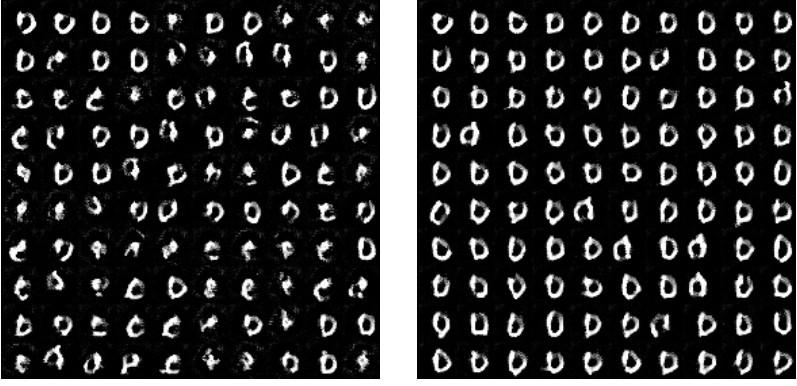

Figure 12: In this instance, the source image, denoted as $\mathbf{x}_{ori}$, represents the digit '0' as displayed in Figure 1, while the target is class 1. We derived 4641 samples from $p_{adv}(\cdot; 0, 1)$ via rejection sampling. The **Left** portion of the figure shows the initial 100 samples, ordered by energy. The **Right** section, on the other hand, depicts the same initial 100 samples, also sorted by energy, but only after retaining the top 10 percent of samples with the highest softmax probability of class 0 in the auxiliary classifier.

## K  PGD ATTACKS ON ADVERSARIALLY TRAINED VICTIM CLASSIFIERS

This section is dedicated to showcasing the application of Projected Gradient Descent (PGD) attacks on robustly trained classifiers, employing a variety of parameters.

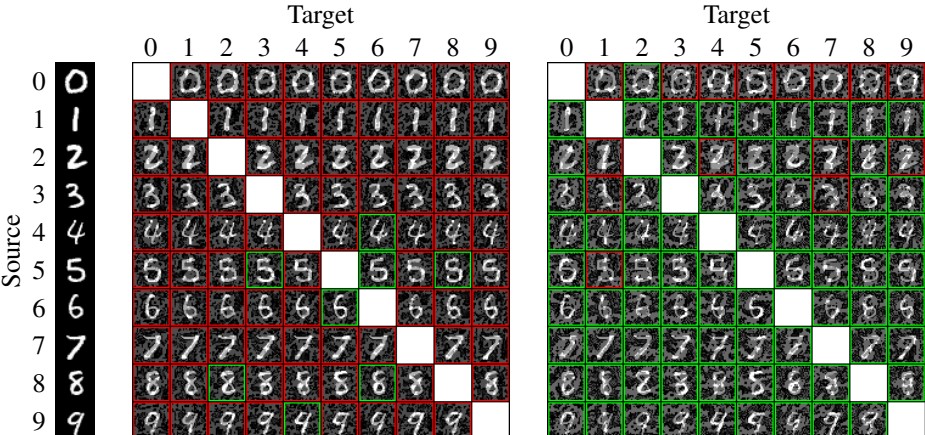

Figure 13: Targeted attacks on an adversarially trained MadryNet (Madry et al., 2017) using Projected Gradient Descent (PGD) with $L_\infty$ norm, $\alpha = 0.04$, and 100 steps. Left: $\epsilon = 0.3$. Right: $\epsilon = 0.4$.

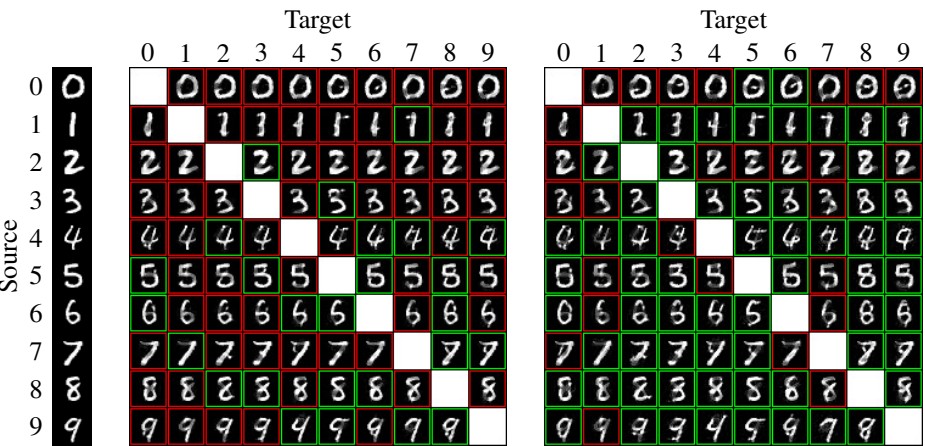

Figure 14: Targeted attacks on an adversarially trained MadryNet (Madry et al., 2017) using Projected Gradient Descent (PGD) with $L_2$ norm, $\alpha = 0.2$, and 100 steps. Left: $\epsilon = 3$. Right: $\epsilon = 4$.

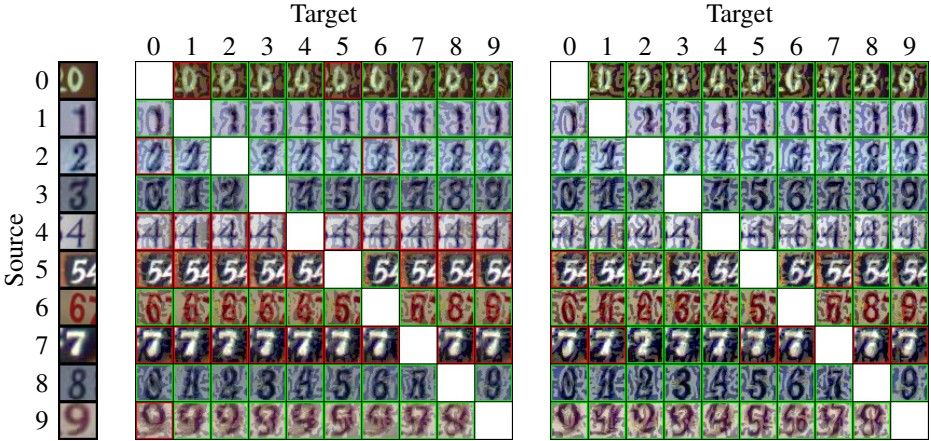

Figure 15: Targeted attacks on an adversarially trained ResNet18 (He et al., 2016) using Projected Gradient Descent (PGD) with $L_\infty$ norm, $\alpha = 0.005$, and 100 steps. Left: $\epsilon = 0.1$. Right: $\epsilon = 0.15$.

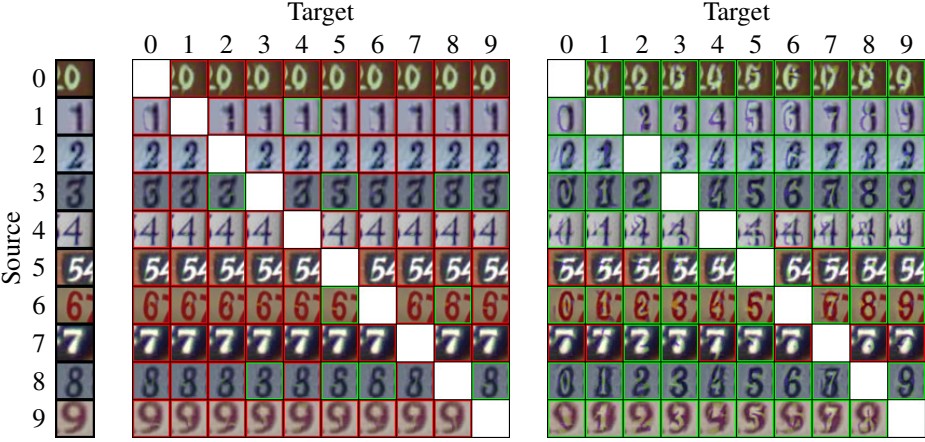

Figure 16: Targeted attacks on an adversarially trained MadryNet (Madry et al., 2017) using Projected Gradient Descent (PGD) with $L_2$ norm, $\alpha = 0.1$, and 100 steps. Left: $\epsilon = 1$. Right: $\epsilon = 3$.

## L  UNRESTRICTED ADVERSARIAL ATTACK

Figure 17 illustrates the unrestricted adversarial examples generated by our method.

## M  ADDITIONAL DISCUSSION ABOUT RELATED WORKS

Xiao et al. (2018) introduced spatially transformed adversarial examples, a type of unrestricted adversarial examples. These are not bounded by geometric distance but rather by the total variation of the flow field. Bhattad et al. (2019) proposed unrestricted perturbations that alter semantically meaningful visual descriptors—color and texture—to create effective and photorealistic adversarial examples. Hosseini & Poovendran (2018) demonstrated that manipulating the HSV color space can produce unrestricted adversarial examples capable of misleading classifiers not trained for adversarial resistance. Additionally, Joshi et al. (2019) developed a GAN-based method to generate adversarial examples by perturbing the latent space.

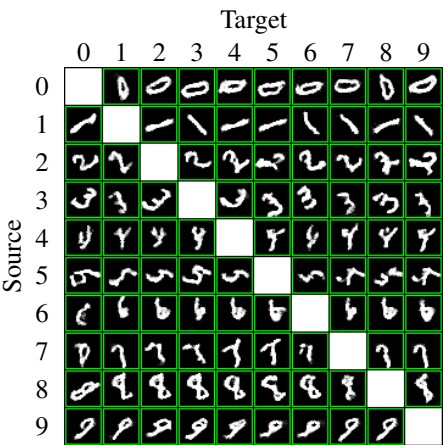

Figure 17: Unrestricted adversarial attack generated by our method

# N    EXPERIMENTS ON RESTRICTED IMAGENET128

As highlighted by Tsipras et al. (2018), adversarial training on the ImageNet dataset is particularly challenging due to the inherent complexity of the classification task and the computational demands of standard classifiers. In line with Tsipras et al. (2018)'s approach, we concentrate on a smaller subset of the dataset. We have grouped semantically similar ImageNet classes into 9 super-classes, detailed in Table 4. Our training and evaluation are exclusively based on examples from these classes.

As currently implemented, EBMs (Du & Mordatch, 2019) do not support high-resolution inputs; therefore, we downscale ImageNet images from their typical size of $224 \times 224$ pixels to $128 \times 128$. Our empirical results suggest that this resolution constraint is a limitation of the current EBM implementations, which we discussed in Section 8.

To reduce the distortion induced by TPS transformations, we employ the original image as the starting point for the Langevin Monte Carlo process. Surprisingly, this method yields significantly better results than the random noise initialization previously utilized. The visual outcomes are illustrated in Figures 18, 19, and 20. By comparing these three images, we can discern that our method leaves fewer traces of tampering.

| Class | Corresponding ImageNet Classes |
|---|---|
| Dog | 151 to 268 |
| Cat | 281 to 285 |
| Frog | 30 to 32 |
| Turtle | 33 to 37 |
| Bird | 80 to 100 |
| Primate | 365 to 382 |
| Fish | 389 to 397 |
| Crab | 118 to 121 |
| Insect | 300 to 319 |

Table 4: Classes used in the Restricted ImageNet128 model. The class ranges are inclusive.

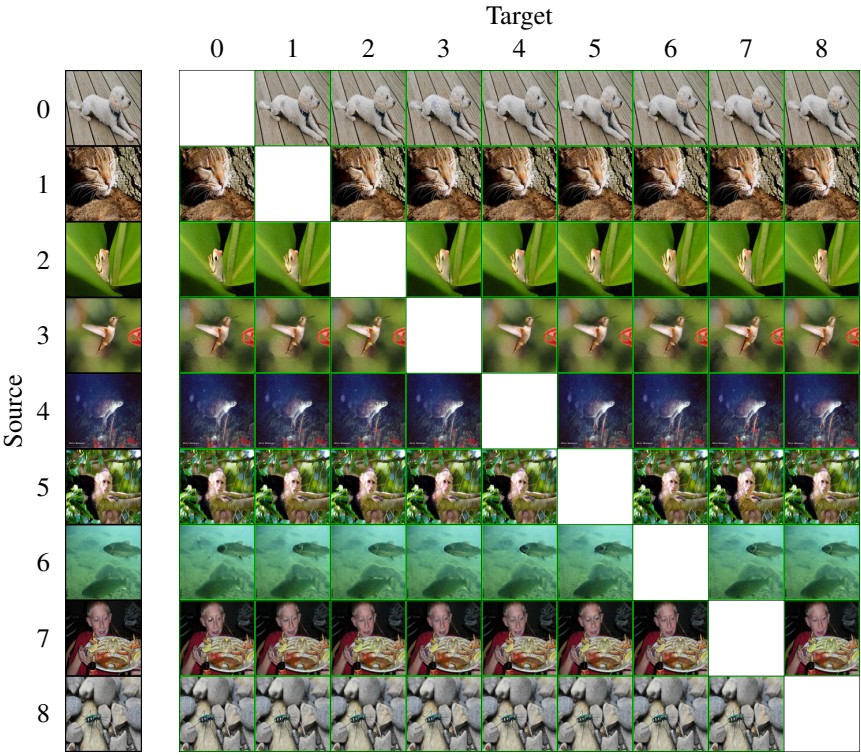

Figure 18: Attack restricted Imagenet128 by our method.

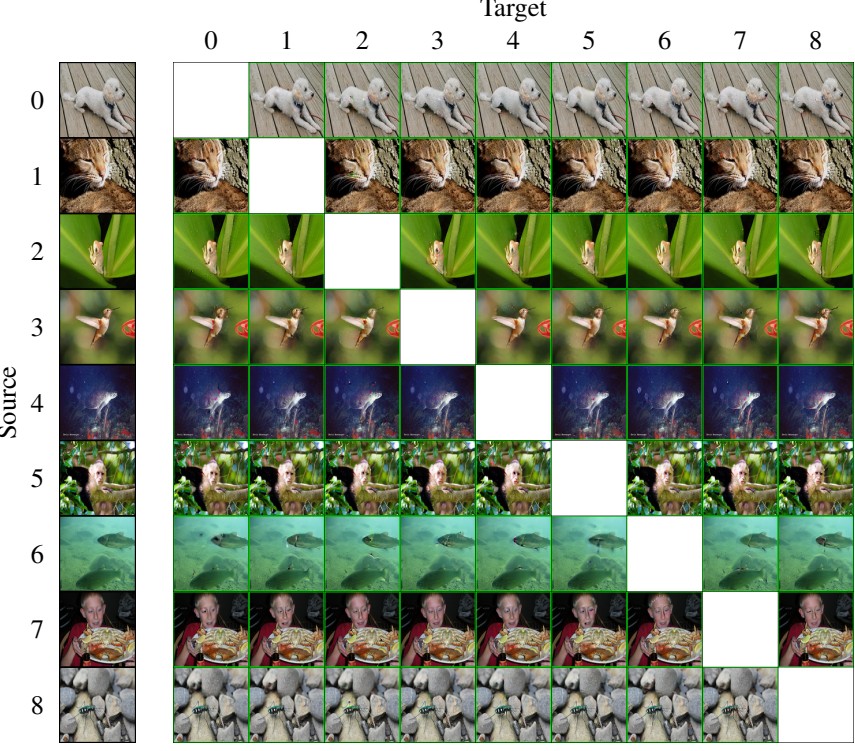

Figure 19: Attack restricted Imagenet128 by PGD with L2 norm.

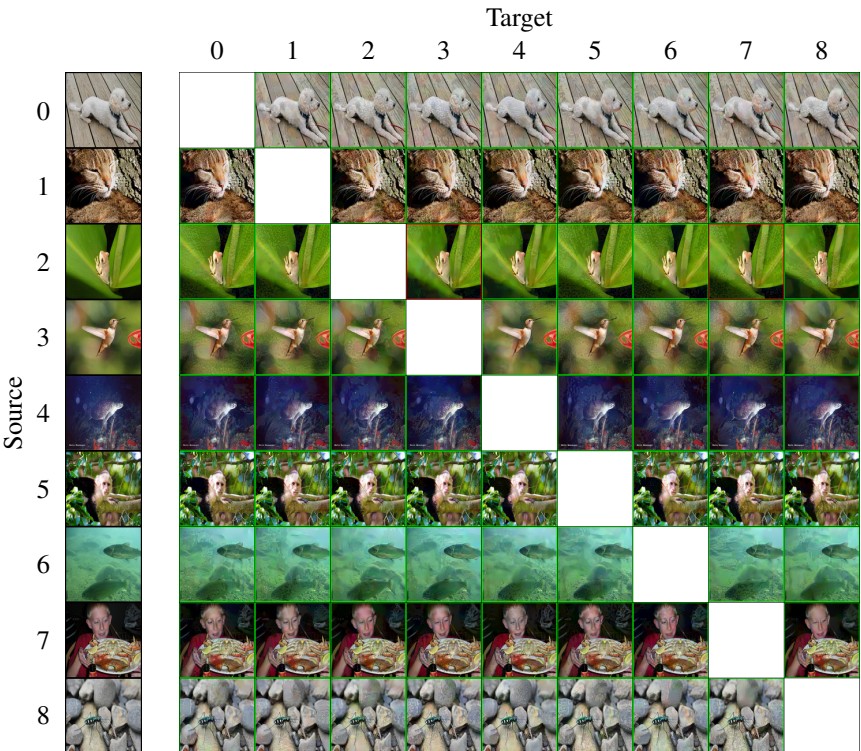

Figure 20: Attack restricted Imagenet128 by PGD with Linf norm.

## O  PROBABILISTIC CW

When $f$ is set as $f_{CW}$ and $p_{dis}$ is chosen to be the Gaussian distribution corresponding to the $L_2$ distance, this configuration results in the probabilistic CW attack ($L_2$). In this scenario, the primary difference between our approach and the traditional CW attack is the optimizer used. Specifically, we utilize Langevin Dynamics for optimization, following the method described by Welling & Teh (2011). Figure 21 demonstrates the outcomes of the Probabilistic CW attack on an adversarially trained MadryNet under various parameters.

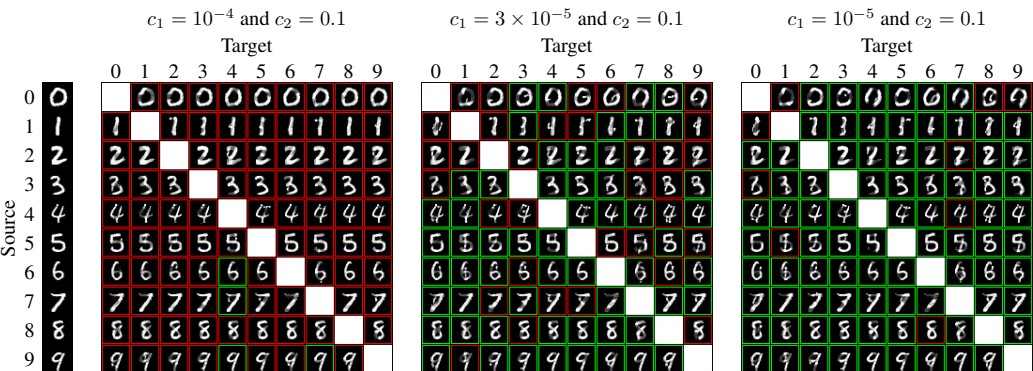

Figure 21: Probablistic CW attack ($L_2$) on MadryNet with adversarial training.

## P  REPLICATION OF STADV

We replicated the StAdv approach as proposed by Xiao et al. (2018), with the results showcased in Figure 22. We employed StAdv to target MadryNet, which was adversarially trained using PGD,

mirroring the settings in our current work. The original paper sets the default value of $\tau$ at $0.05$ for normal classifiers. However, we found this setting inadequate for adversarially trained classifiers. Consequently, we adjusted the $\tau$ parameter to more effectively deceive the adversarially trained MadryNet in most instances, setting $\tau$ to $0.001$.

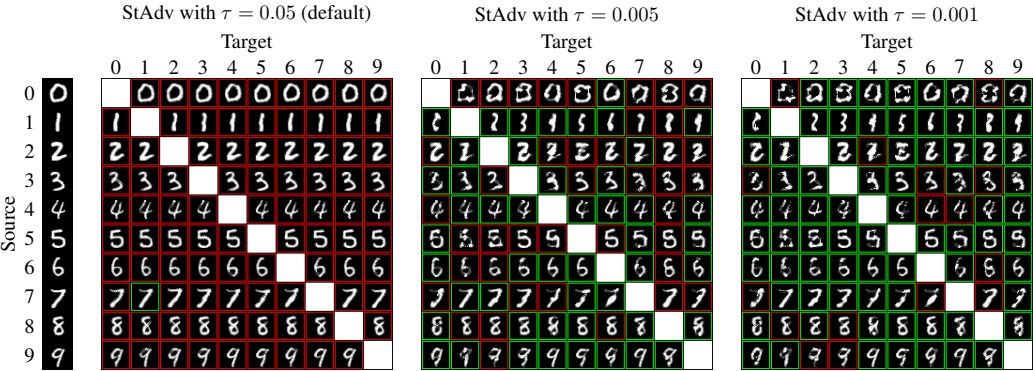

Figure 22: StAdv Attack on MadryNet with adversarial training.

## Q    COMPARISON OF PROB CW, STADV AND OUR PROPOSED METHOD

Figure 23 showcases adversarial examples generated by Prob CW, StAdv, and our proposed method, each attacking the same adversarially trained MadryNet. The examples reveal that Prob CW and StAdv's outputs exhibit noticeable tampering. For instance, when '0' is the source, Prob CW's adversarial examples often fail to form a complete circle, whereas most of StAdv's examples, though complete, lose the circular shape. In contrast, our method maintains the zero shape. Similarly, when '1' is the source, both Prob CW and StAdv produce examples that take on the target class's semantics, indicating a failure. In various instances, our method exhibits notable superiority over these alternatives. For example, many adversarial examples generated by their methods display shadows, a clear indication of tampering. This visual superiority underscores the benefits of our data-driven $p_{\text{dis}}$.

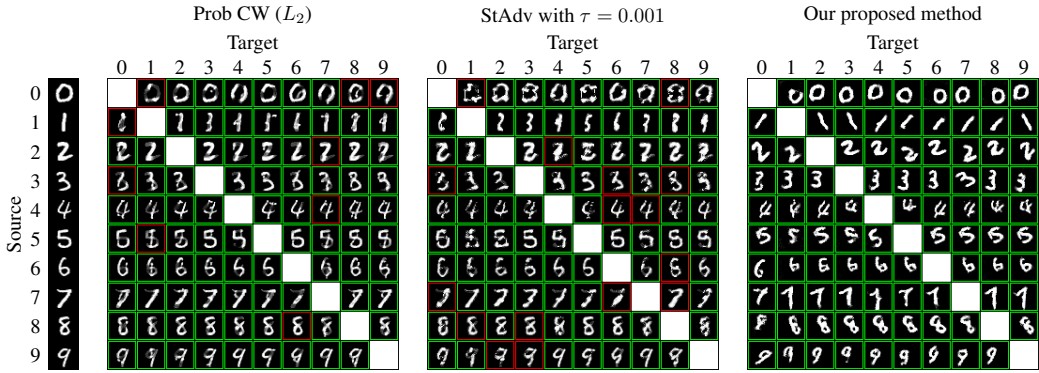

Figure 23: Comparison of Prob CW, StAdv and Our Proposed Method

## R    BROADER IMPACTS OF THIS WORK

The present study introduces a novel approach: the semantics-aware adversarial attack. This method provides significant insights into the resilience and vulnerability of sophisticated classifiers.

From an advantageous perspective, it highlights the inherent risks associated with robust classifiers. By exposing potential weak points in such systems, the study underscores the necessity for further

improvements in classifier security. This can pave the way for building more resilient artificial intelligence systems in the future.

Conversely, the work also presents potential pitfalls. There is a risk that malicious entities might exploit the concepts discussed here for nefarious purposes. It is crucial to take into account the potential misuse of this semantics-aware adversarial attack and accordingly develop preventive measures to deter its utilization for unethical ends.

