# OpenReview forum: "Constructing Semantics-Aware Adversarial Examples with Probabilistic Perspective"
_ICLR.cc/2024/Conference — Submitted to ICLR 2024_

### Official Review · Reviewer_sPYy · 2023-10-27

**Soundness:** 3 good
**Presentation:** 3 good
**Contribution:** 3 good
**Rating:** 6
**Confidence:** 2

**Summary:**

This paper aims to generate adversarial examples while avoid compromising semantic information. To achieve this goal, they propose a semantic distance which replaces the geometrical distance. Then, they utilize the technique of Langevin Monte Carlo to search the adversarial examples. Instead of generate adversarial examples in a geometrical constraint, they transition to a trainable, data-driven distance distribution, which can incorporate personal comprehension of semantics into the model. The generated adversarial examples shown in Experiment seem more natural and untouched than the tradition PGD-generated examples.

**Strengths:**

* The problem is interesting to me. Although adversarial examples are widely known as imperceptible to human eyes, some adversarial examples are vague compared to the original images. This work generates more clear adversarial examples, which are harder for humans to detect.
* The algorithm is theoretically supported and empirically efficient.

**Weaknesses:**

* Although the generated adversarial examples are indeed elusive to human eyes, it is unknown whether these examples are harder for the defense methods to detect.
* It would be better if the authors can also validate the efficacy of the proposed method on larger network such as ResNet-50, and more difficult task such as CIFAR-100 and ImageNet.

**Questions:**

What is the perturbation radius used in Figure 1?

---

> ### Author Response · Authors · 2023-11-13
> **Response to Reviewer sPYy**
>
> Thank you for taking the time to review our work. Here is our response:
>
> > some adversarial examples are vague compared to the original images.
>
> Yes, indeed, this phenomenon is particularly evident in adversarially trained classifiers, which are notably challenging to attack.
>
> > ..., it is unknown whether these examples are harder for the defense methods to detect.
>
> Through reject sampling and sample refinement, the samples presented to human annotators are already capable of deceiving the victim classifier. Due to the length constraints of our submission, we detailed reject sampling and sample refinement in Appendix D, which might have led to some misunderstanding. We apologize for any confusion this may have caused.
>
> > It would be better if the authors can also validate the efficacy of the proposed method on larger network such as ResNet-50, and more difficult task such as CIFAR-100 and ImageNet.
>
> We have observed a notable decline in classification performance in multi-class classifiers with a large number of classes after adversarial training, and the improvement in defense is not substantial. Consequently, we did not select CIFAR-100 as our target, following the precedent set by works [1] and [2]. To showcase the capability of our method with larger images, we are currently experimenting with ResNet-50 on a restricted version of ImageNet, aligning with the methodology in [1]. Updates on the progress of this additional experiment will be provided as soon as it is complete.
>
> > What is the perturbation radius used in Figure 1?
>
> In Figure 1, the perturbation radius ($\epsilon$) for PGD with $L_{\infty}$ norm are set to 0.4, 0.15, and 0.1 for MNIST, SVHN, and CIFAR10, respectively. For PGD with the $L_2$ norm, the corresponding $\epsilon$ values are 5, 4, and 3 for the same datasets.
>
> [1] Tsipras, Dimitris, et al. "Robustness may be at odds with accuracy." arXiv preprint arXiv:1805.12152 (2018).
>
> [2] Madry, Aleksander, et al. "Towards deep learning models resistant to adversarial attacks." arXiv preprint arXiv:1706.06083 (2017).

---

> ### Author Response · Authors · 2023-11-21
> **ImageNet Update**
>
> The results of the ImageNet experiment have been compiled. Kindly refer to Appendix N in the present version of the PDF, located at the document's end. As I am in the process of revising the document, I will inform you should there be any changes to this reference.
>
> Please review the ImageNet results at your earliest convenience and do not hesitate to reach out with any questions you might have. Thank you!

---

### Official Review · Reviewer_DtN5 · 2023-10-28

**Soundness:** 2 fair
**Presentation:** 3 good
**Contribution:** 2 fair
**Rating:** 5
**Confidence:** 3

**Summary:**

This paper introduces generating adversarial examples from a probabilistic perspective. The geometric constraints of adversarial examples are interpreted as distributions, thus facilitating the transition from geometric constraints to data-driven semantic constraints. The paper introduces relevant background knowledge in detail and introduces four techniques that enhance the performance of our proposed method in generating high-quality adversarial examples.  The effectiveness of the proposed method was verified on some simple dataset.

**Strengths:**

1. It provides detailed background knowledge on paper writing and is very reader-friendly.
2. The author models adversarial examples from a probabilistic perspective and proposes an energy model-based adversarial example generation method.

**Weaknesses:**

1.  Although the author models the generation process of adversarial examples through the energy model, there is no essential difference between the previous adversarial example generation processes (they all use gradients to find optimization targets，the objective function used in this paper is still the C&W attack.).
2. The method has only been verified on some small datasets( e.g., MNIST and CIFAR), and the effectiveness of the method needs to be verified on larger-scale and more complex semantic datasets(e.g., ImageNet). Although the authors discuss this weaknesses.

**Questions:**

Some questions：
1:  Generating adversarial samples through the energy model requires multiple data transformations. The author discussed changes in semantic distribution and geometric distribution. Can defenders train a classifier to filter adversarial samples generated based on the energy model?

---

> ### Author Response · Authors · 2023-11-12
> **Response to Reviewer DtN5**
>
> Thank you for your review. Below is our response:
>
> > It provides detailed background knowledge on paper writing and is very reader-friendly.
>
> I am greatly pleased to hear your acknowledgment. We have put effort into making this section clear, as our goal is to bridge the gap in understanding between those focused on probabilistic modeling and those specializing in adversarial attack applications.
>
> > there is no essential difference between the previous adversarial example generation processes (they all use gradients to find optimization targets，the objective function used in this paper is still the C&W attack.).
>
> We respectfully disagree with this point. As detailed in formula (1), the objective function comprises two components: $\mathcal{D}$ and $f$. While it's true that we selected $f_{CW}$ as a common choice for $f$, the unique contribution of our probabilistic approach lies in the data-driven aspect of $\mathcal{D}$. This means that $\mathcal{D}$ is not just a geometric distance anymore. The distance distribution $p_{\text{dis}}$ can be modeled by any probabilistic generative model. In our case, we found that the Energy-Based Model (EBM) provides an elegant mathematical formulation, allowing us to use minus energy to represent distance. We believe we have already highlighted this point in Section 4 of our original submission.
>
> Moreover, we believe that the version you mentioned is identified as the 'probabilistic CW attack' (prob CW), wherein $\mathcal{D}$ is the L2 norm and $f$ aligns with $f_{CW}$, as depicted in Figure 3 (c). The primary distinction between prob CW and the conventional CW attack is the implementation of Langevin dynamics for the optimization process.
>
> > The method has only been verified on some small datasets.
>
> We have tested our proposed method on the MNIST, SVHN, and CIFAR10 datasets, which is consistent with the experimental scale in [1]. As highlighted in Section 8, attacking digit datasets, despite their lower resolution, is practically more challenging. We believe the scope of our current experiments sufficiently demonstrates the strengths of our method. The primary limitation in extending to higher resolution images stems from the capabilities of the Energy-Based Model (EBM), which we used as one possible representation of $ p_{\text{dis}}$. It's plausible that choosing a different $p_{\text{dis}}$ capable of handling higher resolution images could overcome this limitation. In this study, we chose EBM for its elegant mathematical form in the probabilistic context, presenting it as a foundational implementation while considering alternative $p_{\text{dis}}$ for future research.
>
> Nevertheless, we will endeavor to include higher resolution experiments in this rebuttal and will keep you updated with any new experimental results.
>
> > Can defenders train a classifier to filter adversarial samples generated based on the energy model?
>
> Certainly, if defenders know about the set of transformations $\mathcal{T}$ beforehand, they might integrate these into their training as data augmentation. However, our study is focused on victim classifiers that have undergone standard adversarial training, a prevalent method consistent with the approach in [1]. We haven't tested attacks on classifiers trained with such transformations, operating under the assumption that defenders are not privy to the attacker’s specific transformations in advance.
>
> [1] Song, Yang, et al. "Constructing unrestricted adversarial examples with generative models." Advances in Neural Information Processing Systems 31 (2018).

---

> > ### Author Response · Authors · 2023-11-21
> > **ImageNet Update**
> >
> > The results of the ImageNet experiment have been compiled. Kindly refer to Appendix N in the present version of the PDF, located at the document's end. As I am in the process of revising the document, I will inform you should there be any changes to this reference.
> >
> > Please review the ImageNet results at your earliest convenience and do not hesitate to reach out with any questions you might have. Thank you!

---

> > > ### Comment · Reviewer_DtN5 · 2023-11-23
> > > **Reply to author**
> > >
> > > Thanks to the author for the response,  I read the rebuttal and some of my concerns were addressed. However, It is unacceptable that an attack method currently cannot be applied to large-resolution images, especially in 2023.
> > > In the supplementary experiment, the author still only conducted experiments at 128 resolution, and did not provide the attack success rate. The author showed some attacked images, which was not convincing.
> > >
> > >
> > > I keep my rating!

---

> ### Author Response · Authors · 2023-11-23
>
> Thanks for your reply!
>
> > However, It is unacceptable that an attack method currently cannot be applied to large-resolution images, especially in 2023.
>
> My understanding is that the primary goal of academic conferences is to propose and exchange innovative ideas, rather than to engage in benchmarking or competition. I encourage you to focus on the originality and innovation of our ideas. It's unreasonable to expect the first steam locomotive to outperform horse-drawn carriages in every performance aspect. Similarly, a data-driven $p_{dis}$ represents a novel concept with considerable flexibility and future potential, particularly as probabilistic generative models continue to evolve.
>
> Furthermore, considering that the popular resolution for ImageNet classification is 224x224, we believe that our resolution of 128x128 is comparable in scale.
>
> > ... and did not provide the attack success rate. The author showed some attacked images, which was not convincing.
>
> As previously mentioned, all the images generated by our method successfully deceive the classifier, achieving a 100% success rate. However, the challenge with unrestricted adversarial attacks is that they may be perceptible to human observers, as they are not assessed under an objective geometric distance. Therefore, a human subjective evaluation becomes crucial. We recognize, as demonstrated in Appendix N, that while our attack method generates adversarial examples with less noticeable tampering traces, the images tend to be relatively blurred. This blurring effect could be attributed to the current limitations in the implementation of energy-based models.

---

### Official Review · Reviewer_URQr · 2023-10-31

**Soundness:** 2 fair
**Presentation:** 3 good
**Contribution:** 1 poor
**Rating:** 5
**Confidence:** 5

**Summary:**

The paper presents a probabilistic approach to construct semantically meaningful adversarial examples by interpreting geometric perturbations as distributions. The approach relies on training an energy based model (EBM) to emulate such distributions. The approach is evaluated on a variety of image classification datasets.

**Strengths:**

1. The proposed approach is a principled take on the semantic adversarial attacks where in the authors derive an EBM-style formulation to generate geometric adversarial examples.

2. The algorithm presents high attack success rates on adversarially trained models.

**Weaknesses:**

1. The paper is missing several references, and is relatively sparse with regards to related work. Similar works include Spatially Transformed Adversarial Examples (Xiao et al., ICLR 2018), Semantic Adversarial Attacks (Joshi et al, 2019, ICCV 2019), Semantic Adversarial examples (Hosseini et al, CVPRW, 2018) and Unrestricted adversarial examples (Bhattad et al, ICLR 2020). Several of these works use generative models to generate semantic adversarial examples and should be discussed and contrasted in this work.

2. The experiments are limited to just two adversarially trained networks. Furthermore, the adversarially trained models do not appear to have been trained under the semantic attack threat model and hence unlikely to be robust to such attacks.

3. The attack also relies on adversarially trained models learning semantically meaningful features. However, the images shown in the figures appear to be distorted which could be a consequence of the models overfitting on a subset of semantic features.

**Update**: The authors have addressed some of my concerns and clarified certain points of misunderstanding. I am therefore increasing my score to 5 to reflect this.

**Questions:**

Could the authors clarify the training setup for the adverarially trained models?

---

> ### Author Response · Authors · 2023-11-11
> **Response to Reviewer URQr**
>
> Thank you for taking the time to review my submission. Below are our detailed responses to your comments.
>
> > The paper presents a probabilistic approach to construct semantically meaningful adversarial examples by interpreting geometric perturbations as distributions. The approach relies on training an energy based model (EBM) to emulate such distributions.
>
> In addition to the concrete approach, we also want to highlight the significance of our probabilistic perspective presented in Section 3. The distance distribution $p_{dis}$ can be modeled using various probabilistic generative models. This probabilistic framework holds considerable potential for extension, with Energy-Based Models (EBM) being just one of many possible options. In this paper, we opted for EBM because its mathematical formulation aligns perfectly with Langevin dynamics, resulting in clear and concise derivations.
>
> > The paper is missing several references, and is relatively sparse with regards to related work. Similar works include ...
>
> We contend that, with the probabilistic modeling perspective, the works cited are not similar to our approach. While some of these studies may employ Generative Adversarial Networks (GANs), it's important to note that GANs do not explicitly model the probability distribution $p(x)$. Our work is pioneering in rigorously deriving the distribution $p_{adv}$ from which adversarial samples are generated.
>
> We acknowledge that from an application perspective, these works bear similarities to ours. In fact, under this viewpoint, any study not employing geometric constraints on the original image could be considered similar to our approach. However, our focus is on the novelty of the probabilistic perspective and modelling. Our contribution lies in introducing new concepts to the academic community, rather than solely advancing application benchmarks.
>
> Thank you for introducing these related works to us. We will ensure to incorporate them into the revised version of our paper.
>
> > The experiments are limited to just two adversarially trained networks.
>
> In the main text, we discuss three adversarially trained networks: two are presented in Table 1 for MNIST and SVHN, and another is detailed in Section 6.2 for CIFAR10. Furthermore, Appendix F features 12 additional networks, demonstrating our model's transferability. We limited the number of structures in the main text for two reasons: firstly, to maintain alignment with the settings in [1], and secondly, to efficiently illustrate our method's advantages without unnecessarily utilizing human annotators.
>
> > Furthermore, the adversarially trained models do not appear to have been trained under the semantic attack threat model and hence unlikely to be robust to such attacks.
>
> The objective of our experiment is to demonstrate the effectiveness of our attack method against adversarially trained models. While some studies in this field have incorporated experiments where generated adversarial examples are used in training robust classifiers, we have not pursued this approach. We believe our current experimental setup sufficiently highlights the strengths of our attack method, in line with [1] and [3], and we aim to avoid unnecessary use of human annotators.
>
> > The attack also relies on adversarially trained models learning semantically meaningful features.
>
> This is not correct. I encourage a careful review of the draft, particularly Figure 2 (a) and (b). These figures demonstrate that when the victim classifier is adversarially trained, $p_{vic}$ possesses generative capabilities, meaning it tends to produce images resembling those from the target class, which in turn makes an adversarial attack more challenging. Our proposed attack method is not dependent on adversarially trained models. In fact, attacking a victim classifier that is not adversarially trained would be significantly easier.
>
> > However, the images shown in the figures appear to be distorted which could be a consequence of the models overfitting on a subset of semantic features.
>
> Could you please specify which distortion you're referring to and in which figure it occurs?
>
> > Could the authors clarify the training setup for the adverarially trained models?
>
> We use the setting introduced in [2], as introduced in Section 2.2.
>
> [1] Song, Yang, et al. "Constructing unrestricted adversarial examples with generative models." Advances in Neural Information Processing Systems 31 (2018).
>
> [2] Madry, Aleksander, et al. "Towards deep learning models resistant to adversarial attacks." arXiv preprint arXiv:1706.06083 (2017).
>
> [3] Xiao, Chaowei, et al. "Spatially transformed adversarial examples." arXiv preprint arXiv:1801.02612 (2018).

---

> > ### Comment · Reviewer_URQr · 2023-11-20
> > **Response**
> >
> > I thank the authors for promptly responding to my review and appreciate the detailed response.
> >
> > 1.  Firstly, I appreciate the clarification that the EBMs used to model the geometric distribution can be replaced with any probabilistic generative model. However, the paper only presents results with EBMs and therefore it is hard to evaluate the effectiveness of the attack for better generative models like diffusion or variational autoencoders.
> >
> > 2. In terms of comparisons, while I do agree with the authors that their contribution of explicitly modelling the distribution is important, it is equally important to compare with similar approaches in order to evaluate the proposed attack. An important additional evaluation metric would be to analyse how accurately do EBMs model the adversarial distribution in comparion to GANs or even uniform distributions over the transformations (perhaps by studying the number of adversarial samples). This would provide more support to the theoretical argument.
> >
> > 3. I thank the authors for pointing me to the additional experiments. I suggest including them in the main draft rather than the appendix (by perhaps reducing the background discussion).
> >
> > 4. It is well known that adversarially trained models are not robust to attacks generated under different threat models, (for example [1](https://openreview.net/pdf?id=Sy8WeUJPf).). As such, it would be unfair to evaluate semantic adversarial attacks on models trained to be robust to $\ell_\infty$ attacks. A more principled would be to actually use the geometric adversarial examples generated to train a robust model, and then evaluate the attack to see if it can still find an adversarial distribution under the given threat model.
> >
> > 5. I thank the authors for clarifying my understanding of fig. 2. I see now that the adversarial distribution would have more support in case of non-adversarially trained models.
> >
> > 6. Both Fig 4 and Fig 10 show recognizably different images from the source, and do not look like numbers sourced from MNIST. I referred to these examples as distorted. Specifically almost all images show broken strokes and grey patches which while may be adding adversarial features, do not look like numbers that are part of the MNIST dataset.
> >
> > I again thank the authors and hope to continue the discussion.

---

> > > ### Author Response · Authors · 2023-11-21
> > >
> > > This message is a continuation of my previous reply. **Please note that each text box has its own separate reference list.**
> > >
> > > ### 4
> > >
> > > > It is well known that adversarially trained models are not robust to attacks generated under different threat models,
> > >
> > > We acknowledge that the adversarially trained MadryNet is not entirely efficient against all kinds of attacks. Our choice of MadryNet as the victim classifier in our experiments was to maintain consistency with the settings used in [2].
> > >
> > > >  A more principled would be to actually use the geometric adversarial examples generated to train a robust model, and then evaluate the attack to see if it can still find an adversarial distribution under the given threat model.
> > >
> > > Could it be that the term 'geometric adversarial examples' actually refers to 'semantic adversarial examples'? We acknowledge that this approach is commonly used in adversarial attack research. However, it is not ideal for our unrestricted adversarial example framework. Our method aims to provide a data-driven distance distribution allowing users to incorporate their semantic understanding. This is based on the assumption that the transformation $\mathcal{T}$, introduced in Section 4, is not previously known to the victim classifier.
> > >
> > > By the way, the elastic-net (EAD) attack you mentioned [1] leverages a weighted sum of L1 and L2 distances, which in turn leads to product of a Laplace distribution and a Gaussian distribution for $p_{dis}$. Indeed, by selecting an optimal weight $\beta$, the elastic-net strategy demonstrates effective results in attacking Madrynet, thereby highlighting the good performance of the induced $p_{dis}$. However, the $p_{dis}$ that arises from geometric distance represents just a minuscule portion of all possible $p_{dis}$ choices. We posit that as probabilistic generative models continue to evolve, data-driven $p_{dis}$ models should be able to provide superior performance.
> > >
> > > ### 5
> > >
> > > Yes, exactly. This offers a probabilistic explanation for why classifiers trained on adversarial examples are comparatively more resistant to deception.
> > >
> > > ### 6
> > >
> > > Yes, in this sense, the MNIST classifier is the most challenging to deceive among various dataset classifiers due to the limited diversity of handwritten data. As seen in Figure 3 of [2] and Figure 2 of [3], their MNIST samples exhibit similar features like broken strokes and grey patches. The samples in our Figure 4 and Figure 10 are pre-reject sampling and refining (see Appendix D), which is why they may appear less refined compared to the MNIST samples in [2] and [3].
> > >
> > > ### Reference
> > >
> > > [1] Sharma, Yash, and Pin-Yu Chen. "Attacking the madry defense model with $ l_1 $-based adversarial examples." arXiv preprint arXiv:1710.10733 (2017).
> > >
> > > [2] Song, Yang, et al. "Constructing unrestricted adversarial examples with generative models." Advances in Neural Information Processing Systems 31 (2018).
> > >
> > > [3] Xiao, Chaowei, et al. "Spatially transformed adversarial examples." arXiv preprint arXiv:1801.02612 (2018).

---

> ### Author Response · Authors · 2023-11-21
>
> Thank you for your reply! This response will be divided into two text boxes.
>
> ### 1
>
> In Section 3, we present the abstract form of the probabilistic perspective on adversarial attacks. The $p_{\text{dis}}$ can represent any probabilistic generative model. Given the abstract nature of this framework, it's not essential to exhaustively explore all possible choices for $p_{\text{dis}}$. Instead, we view it as a promising research direction for future exploration. In this work, we use the EBM as a case to demonstrate the framework's potential. Our current draft aims to introduce a new perspective and a model based on it, rather than achieving the ultimate performance in this task.
>
> ### 2
>
> >  it is equally important to compare with similar approaches in order to evaluate the proposed attack.
>
> We agree that comparing with similar approaches is crucial for evaluating our proposed attack. However, aligning comparisons in unrestricted adversarial attacks is challenging. In restricted attacks, we can standardize comparisons by fixing the L-p distance and comparing success rates. For unrestricted attacks, while we can achieve a 100% success rate in deceiving the classifier, it's unclear whether the adversarial examples deviate significantly from the original image in human perception. Therefore, we rely on human annotators to determine if the original class of the image is recognizable. This method, aligned with [1], is reflected in our Table 1 results.
>
> Contrastingly, [2] proposes a method that doesn't guarantee a 100% success rate in deceiving the victim classifier. Consequently, they employ a different evaluation method, assuming their adversarial examples are imperceptible to humans. Their success rate is thus based on the classifier's deception rate. To validate their assumption, they conducted an A/B test assessing human perception, focusing on ImageNet. However, this method may lack rigor, as indicated by the discernibility of their MNIST adversarial samples in Figure 2 of [2], which might be more noticeable to humans than suggested.
>
> [3] similarly assumes that their method produces adversarial examples undetectable by humans and compares the accuracy of the victim classifier, where a lower accuracy indicates a more effective attack. However, upon examining their figures, I find their assumption about human indistinguishability unconvincing.
>
> [4], being a workshop paper, lacks comparative analysis.
>
> [5] assumes that alterations in color and texture remain undetected by humans. Subsequently, they assess their attack's success rate in comparison with conventional methods.
>
> In summary, **our approach and [1], guarantee that our adversarial examples successfully deceive the victim classifier, with human annotators then evaluating their distinguishability. In contrast, [2], [3], and [5] proceed on the assumption that their generated adversarial examples are indistinguishable to humans, focusing instead on comparing the success rates in compromising the victim classifier.**
>
> Therefore, in our submission, we limit our comparison to [1]. In the revised version, we will discuss [2], [3], [4], and [5], but will not include them in our comparison.
>
> > how accurately do EBMs model the adversarial distribution in comparion to GANs
>
> [1]'s approach uses GANs to implicitly model the distribution of adversarial examples, whereas our method explicitly models this distribution. Direct comparison of these distributions is challenging, leading us to select human annotators' success rate as our sole evaluation metric.
>
> > even uniform distributions over the transformations
>
> I may have misunderstood your question, but if we were to use a uniform distribution for $p_{\text{dis}}$, then $p_{\text{adv}}$ would essentially reduce to $p_{\text{vic}}$, as depicted in Figure 2 (a) and (b).
>
> ### 3
>
> Thank you for your suggestion. We will relocate the training and sampling details of the EBM to the Appendix and shift the table showing transferability results to the main text.
>
> ### Reference
>
> [1] Song, Yang, et al. "Constructing unrestricted adversarial examples with generative models." Advances in Neural Information Processing Systems 31 (2018).
>
> [2] Xiao, Chaowei, et al. "Spatially transformed adversarial examples." arXiv preprint arXiv:1801.02612 (2018).
>
> [3] Joshi, Ameya, et al. "Semantic adversarial attacks: Parametric transformations that fool deep classifiers." Proceedings of the IEEE/CVF international conference on computer vision. 2019.
>
> [4] Hosseini, Hossein, and Radha Poovendran. "Semantic adversarial examples." Proceedings of the IEEE Conference on Computer Vision and Pattern Recognition Workshops. 2018.
>
> [5] Bhattad, Anand, et al. "Unrestricted adversarial examples via semantic manipulation." arXiv preprint arXiv:1904.06347 (2019).

---

> ### Comment · Reviewer_URQr · 2023-11-21
> **Thanks for the response**
>
> I appreciate the quick reply and the clarifications as well as the new Imagenet results. However, I still disagree with the authors on comparisons. Perhaps it is harder to exhaustively compare all semantic adversarial methods fairly due to the varying setups. However, [1] provides a very similar threat model. Also, while the authors claim that [1] does not have a 100% success rate and therefore cannot be compared, Figs 3 and 4 from the main paper show similar failures. It might be useful to compare the failure modes to evaluate how semantic divergence is a better approach. In addition, given the significant progress in our understanding of adversarial robustness, and the standard evaluation approaches, evaluating a robust model under an attack with a different threat model does not make sense to me. Perhaps the other reviewers could chime in and present their opinions on this.
>
> However, I do find the idea of learning an adversarial distribution interesting by itself as well as of interest to the community if it can be scaled to larger models and complex datasets. As of now, I am willing to increase my score to 5 (marginally below the acceptance threshold). I am also open to changing my mind if the other reviewers or authors have additional thoughts to support the paper.
>
> [1] Xiao, Chaowei, et al. "Spatially transformed adversarial examples."

---

> > ### Author Response · Authors · 2023-11-22
> > **Thanks for the reply!**
> >
> > Thank you for your prompt response and engaging participation in our discussion!
> >
> > > However, [1] provides a very similar threat model.
> >
> > The distinction between the approach in [1] and our proposed method is substantial. Reference [1] employs a meticulously crafted, ad hoc strategy for generating adversarial examples. In contrast, our method is data-driven, enabling users to integrate their insights on semantic invariance simply by suggesting transformations.
> >
> > I believe you perceive our approach as similar to [1] due to the resemblance between their spatial transformation, defined by flow, and the TPS transformation we illustrate in our paper. Indeed, visually, [1] appears akin to our method when employing TPS as the transformation. However, our framework treats TPS as just one of several transformation options. For instance, in the MNIST dataset, we employ a combination of TPS, translation, scaling, and rotation, as shown in the top right of Figure 1. Similarly, in the SVHN case, we meld TPS with color space transformations, akin to those in [2] and [3].
> >
> > The selection of transformations in our framework is both flexible and subjective, allowing users to choose combinations they believe do not alter the semantics. Nevertheless, this subjectivity may not always align with others' views, a point we have discussed in Appendix C.
> >
> > > ... It might be useful to compare the failure modes to evaluate how semantic divergence is a better approach.
> >
> > Upon thorough review of [1], we observe that Figures 2 and 11 in [1] showcase adversarial examples generated for classifiers without adversarial training. This is further supported by an open-source replication available at https://github.com/as791/stAdv-PyTorch/tree/main (as illustrated in the README of this repository). Notably, [1] does not present adversarial examples for classifiers that have undergone adversarial training. Therefore, their assertions in Table 3, which are based on the assumption that their adversarial examples remain imperceptible to humans even when applied to adversarially trained classifiers, are not rigorously validated. This highlights the superiority of our proposed method, which produces similarly results even when targeting adversarially trained models. If time allows before the discussion deadline, I aim to replicate [1] on an adversarially trained model and include these results in the appendix.
> >
> > > ... evaluating a robust model under an attack with a different threat model does not make sense to me.
> >
> > I am uncertain if the prevailing standard evaluation methods involve using a proposed adversarial attack method to generate adversarial examples, subsequently employing these examples to train a robust model, and then attacking this model with the same adversarial method. While this appears to be an intriguing experiment, I am unsure if it represents a current convention. Could you provide some examples to clarify?
> >
> > >  ... if it can be scaled to larger models and complex datasets.
> >
> > Thank you for acknowledging our probabilistic perspective. We consider this a novel and straightforward idea, aiming to introduce probabilistic modeling to the adversarial attack community in a principle way. While our current support is limited to images up to 128x128 due to the constraints of a specific probabilistic generative model, our proposed abstract method is not inherently restricted. We are confident that our work represents a significant initial step in advancing this new type of attack and worths attention at academic conferences.
> >
> > ### Reference
> >
> > [1] Xiao, Chaowei, et al. "Spatially transformed adversarial examples."
> >
> > [2] Hosseini, Hossein, and Radha Poovendran. "Semantic adversarial examples." Proceedings of the IEEE Conference on Computer Vision and Pattern Recognition Workshops. 2018.
> >
> > [3] Bhattad, Anand, et al. "Unrestricted adversarial examples via semantic manipulation." arXiv preprint arXiv:1904.06347 (2019).
> >
> > [4] Song, Yang, et al. "Constructing unrestricted adversarial examples with generative models." Advances in Neural Information Processing Systems 31 (2018).

---

> ### Author Response · Authors · 2023-11-22
> **StAdv Replicated**
>
> Thank you for your suggestion! We have successfully replicated the StAdv method as proposed by [1], and the detailed results are now included in Appendix O (**Update: now is P**) of the latest version of our document.
>
> [1] Xiao, Chaowei, et al. "Spatially transformed adversarial examples."

---

> > ### Comment · Reviewer_URQr · 2023-11-22
> > **Thanks for the update**
> >
> > Thanks for the update. Could you also share the success rates of stAdv vs your approach?

---

> > > ### Author Response · Authors · 2023-11-22
> > >
> > > Thank you for your quick reply! The success rates for both our method and that of [1] are determined through human annotation. We believe it's unnecessary to conduct a human annotation experiment for [2]. A visual comparison of Figure 1 and Figure 21 reveals that [2]'s method is more significantly impacted by $p_{vic}$, resulting in its adversarial examples partially reflecting the target class's semantics, particularly in cases where the original images are 0, 1, 3, or 6. I plan to include a detailed human observation of these results in the appendix, though not a full human annotation experiment.
> > >
> > > The primary goal of our experimental design and comparative analysis is to validate our assertions. As previously stated, our approach is inherently data-driven and exhibits greater flexibility compared to [2]. Therefore, we deem a detailed comparison involving human participants with [2] to be unnecessary.
> > >
> > > [1] Song, Yang, et al. "Constructing unrestricted adversarial examples with generative models." Advances in Neural Information Processing Systems 31 (2018).
> > >
> > > [2] Xiao, Chaowei, et al. "Spatially transformed adversarial examples."

---

> > > ### Author Response · Authors · 2023-11-23
> > > **Visual Comparison Update**
> > >
> > > Thank you for your suggestion. For a visual comparison between StAdv and our proposed method, please refer to Appendix Q in our revised submission.

---

### Official Review · Reviewer_CGw2 · 2023-10-31

**Soundness:** 2 fair
**Presentation:** 3 good
**Contribution:** 2 fair
**Rating:** 5
**Confidence:** 3

**Summary:**

This paper propose a novel and simple methods for constructing semantic-aware adversarial examples. The proposed frame work use a probabilistic perspective method to generate semantics-aware adversarial examples.  As depicted in the figure of this paper, it seems this attack successfully attack classifiers while retaining the original image's semantic information.

**Strengths:**

* The method for generating semantics-aware adversarial examples is very intuitive.
* This paper is well-written. The problem this paper focuses is important, and the proposed method is interesting.

**Weaknesses:**

Please correct me if I have some misunderstanding of the paper.

1. Since this paper proposes a semantics-aware method, the human evaluation is important to verify the validity of adversarial examples.

2. It lacks a comparison on the number of queries.

3. There is a lack of an ablation experiment. In $p_{adv}$, what will happen if $p_{dis}$ uses L2 or other methods to calculate Figure 1 or Table 1?

**Questions:**

Please see the weakness section.

---

> ### Author Response · Authors · 2023-11-10
> **Response to Reviewer CGw2**
>
> Thank you for your insightful review and for recognizing the contribution of our paper.
>
> Here are our responses to your comments:
>
> > Since this paper proposes a semantics-aware method, the human evaluation is important to verify the validity of adversarial examples.
>
> Yes, human evaluation is crucial. Our experiments conform to the evaluation methods detailed in [1].
>
> > It lacks a comparison on the number of queries.
>
> Could you clarify what is the number of queries you're referring to?
>
> > There is a lack of an ablation experiment. In $p_\{\text{adv}}$, what will happen if $p_\{\text{dis}}$ uses L2 or other methods to calculate Figure 1 or Table 1?
>
> Refer to Figure 3 (c), showcasing the Prob CW (probabilistic CW attack).
>
> Figure 3 (c) reveals that the adversarial examples represent intersections between the original image and the target class, making them easily identifiable by humans. Therefore, we omitted them from Table 1 to conserve human annotation efforts. In response to your suggestion, we will incorporate adversarial examples generated by the prob CW attack in the format of Figure 1 into our revised draft and will notify you once it's complete.
>
> [1] Song, Yang, et al. "Constructing unrestricted adversarial examples with generative models." Advances in Neural Information Processing Systems 31 (2018).

---

> > ### Author Response · Authors · 2023-11-23
> > **Prob CW update**
> >
> > For a detailed illustration of the Prob CW attack, please see Appendix O and Q in our revised submission.

---

### Author Response · Authors · 2023-11-21
**Imagenet Result**

The results of the ImageNet experiment have been compiled. Kindly refer to Appendix N in the present version of the PDF, located at the document's end. As I am in the process of revising the document, I will inform you should there be any changes to this reference.

Please review the ImageNet results at your earliest convenience and do not hesitate to reach out with any questions you might have. Thank you!

---

### Author Response · Authors · 2023-11-23
**Changes in the Revised Version Compared to the Original Submission**

In the revised version, following Reviewer URQr's suggestion, we have relocated the section on the training and sampling of the energy-based model to an appendix, while moving the evaluation of transferability into the main text.

Additionally, Appendices M, N, O, P, and Q contain new content that addresses the concerns raised by the reviewers.

---

### Author Response · Authors · 2023-11-23
**About originality and evaluation**

Regrettably, our work did not receive a high score after the discussion period. However, we don't believe this should deter the wider community from recognizing the novelty of our approach. As we mentioned in the paper, the inability to support larger resolutions is a limitation of the Energy-based model itself, not a restriction brought about by our novel method. With the advancement of deep learning and the improvement of implementation, we foresee that Energy-based models that support higher resolutions will become available. The most significant contribution of our article lies in introducing a novel probabilistic perspective on adversarial attacks. Under this perspective, we naturally view geometric distance constraints as a distribution, and we can freely replace this with a data-driven distribution. This data-driven approach also provides an avenue for users to incorporate their subjective understanding of semantics. Such an idea might very well become a milestone in the field of adversarial attacks. We believe our work is a good initial step to introduce this new type of attack.

We'd like to stress that neither MNIST nor SVHN datasets should be considered "easy". As demonstrated in our submission and subsequent rebuttal, adversarially trained classifiers for these datasets are more challenging to deceive than those for natural images. This could be attributed to the limited diversity of these digit datasets, allowing robust classifiers to deeply grasp the semantics of each class. As deep learning evolves, we anticipate the emergence of even more semantically-aware classifiers. Our semantics-aware adversarial attack method is well-suited to address this evolution.

Evaluation is crucial but comes with inherent limitations in the progression of science and technology. An overemphasis on evaluation may narrow the focus to specific models like BERT, potentially overlooking the benefits of exploring alternatives such as GPT. Unrestricted adversarial examples, by their very nature of being unrestricted, offer more flexibility, making it challenging to establish a unified benchmark. In other words, the existence of a benchmark often indicates a well-explored area. We believe our experiments adequately support our novel concept. Our goal isn't to outperform others; research is about exploration, not competition. Just as it's unrealistic to expect the first steam locomotive to surpass horse-drawn carriages in all performance aspects.

Moreover, it's not entirely fair to compare our abstract, flexible method with more concrete approaches. However, we do demonstrate instances where our abstract method outperforms others. The true essence of our work – the adaptability of data-driven $p_{dis}$ – cannot be fully captured through direct comparisons.

We appreciate the active participation and constructive advice from reviewer URQr during the discussion. Our thanks also go to all the reviewers and participants for dedicating their time to reviewing this paper.

---

### Meta-Review · Area_Chair_mtzK · 2023-12-16

**Metareview:**

**Scientific Claims and Findings**:
The paper proposes a novel method for generating semantic-aware adversarial examples by introducing a probabilistic perspective. The approach involves interpreting geometric perturbations as distributions and training an energy-based model (EBM) to emulate these distributions. The generated adversarial examples aim to compromise classifiers while preserving the original image's semantic information. The method is evaluated on image classification datasets, and the results suggest successful attacks on adversarially trained models.

**Strengths of the Paper**:

- Intuitive Method: The paper introduces an intuitive method for generating semantics-aware adversarial examples, offering a new perspective on adversarial attacks.
- Clarity and Well-Written: The paper is well-written, and the proposed method is presented in a clear and understandable manner. The problem addressed is important, and the proposed approach is interesting.
- Principled Approach: The approach is principled, relying on a probabilistic formulation and leveraging an energy-based model. The high attack success rates on adversarially trained models indicate the effectiveness of the proposed method.

**Weaknesses and Missing Elements**:

- Lack of Human Evaluation: Since the proposed method aims to be semantics-aware, the paper lacks human evaluation to verify the validity of adversarial examples. Human perception of the generated examples is crucial for assessing the success of the semantic preservation.
- Limited Comparisons and Experiments: The paper lacks a comparison on the number of queries needed for the proposed method. Additionally, there is a lack of an ablation experiment, especially regarding the use of different distance metrics, such as L2. A more comprehensive analysis of the proposed method's sensitivity to various parameters and choices would strengthen the paper.
- Sparse Related Work Discussion: The paper is criticized for missing references and having a relatively sparse discussion of related work. The reviewers suggest the inclusion of references to works like Spatially Transformed Adversarial Examples, Semantic Adversarial Attacks, and Unrestricted Adversarial Examples, which would provide a more comprehensive context for the proposed method.

**Justification For Why Not Higher Score:**

The recommendation for the current score is based on a few key considerations:

- Lack of Human Evaluation: The proposed method aims to generate semantics-aware adversarial examples, which necessitates a human evaluation to verify the validity of these examples. The absence of human perception studies limits the understanding of how well the generated adversarial examples preserve semantic information from a human perspective.

- Limited Comparisons and Experiments: The paper lacks a comparison on the number of queries needed for the proposed method. Additionally, there is a lack of an ablation experiment, especially regarding the use of different distance metrics, such as L2. A more comprehensive analysis of the proposed method's sensitivity to various parameters and choices would provide a more thorough understanding of its effectiveness.

- Sparse Related Work Discussion: The paper is criticized for missing references and having a relatively sparse discussion of related work. Inclusion of references to works like Spatially Transformed Adversarial Examples, Semantic Adversarial Attacks, and Unrestricted Adversarial Examples would enhance the completeness of the related work section and provide a better context for the proposed method.

While the paper introduces an intuitive method with a principled approach and reports promising results, addressing the mentioned limitations would contribute to a more comprehensive evaluation of the proposed method, potentially justifying a higher score.

**Justification For Why Not Lower Score:**

N/A

---

### Decision · Program_Chairs · 2024-01-16

Reject